# Mid-Infrared Optoelectronic Devices Based on Two-Dimensional Materials beyond Graphene: Status and Trends

**DOI:** 10.3390/nano12132260

**Published:** 2022-07-01

**Authors:** Rui Cao, Sidi Fan, Peng Yin, Chunyang Ma, Yonghong Zeng, Huide Wang, Karim Khan, Swelm Wageh, Ahmed A. Al-Ghamd, Ayesha Khan Tareen, Abdullah G. Al-Sehemi, Zhe Shi, Jing Xiao, Han Zhang

**Affiliations:** 1Institute of Microscale Optoelectronics, International Collaborative Laboratory of 2D Materials for Optoelectronics Science and Technology, College of Physics and Optoelectronic Engineering, Shenzhen University, Shenzhen 518060, China; caorui808@163.com (R.C.); fansidicity@163.com (S.F.); zengyonghong777@163.com (Y.Z.); wanghuide@szu.edu.cn (H.W.); karim_khan_niazi@yahoo.com (K.K.); hzhang@szu.edu.cn (H.Z.); 2College of Photoelectrical Engineering, Changchun University of Science and Technology, Changchun 130022, China; yinpengjiayou@163.com; 3Research Center of Circuits and Systems, Peng Cheng Laboratory (PCL), Shenzhen 518055, China; macy@szu.edu.cn; 4Department of Physics, Faculty of Science, King Abdulaziz University, Jeddah 21589, Saudi Arabia; wswelm@kau.edu.sa (S.W.); agamdi@kau.edu.sa (A.A.A.-G.); 5School of Mechanical Engineering, Dongguan University of Technology, Dongguan 523808, China; chemistayesha@yahoo.com; 6Research Center for Advanced Materials Science (RCAMS), King Khalid University, Abha 61413, Saudi Arabia; agsehemi@kku.edu.sa; 7School of Physics & New Energy, Xuzhou University of Technology, Xuzhou 221018, China; 8College of Physics and Electronic Engineering, Taishan University, Tai’an 271000, China

**Keywords:** two-dimensional materials, mid-infrared, modulator, photodetectors

## Abstract

Since atomically thin two-dimensional (2D) graphene was successfully synthesized in 2004, it has garnered considerable interest due to its advanced properties. However, the weak optical absorption and zero bandgap strictly limit its further development in optoelectronic applications. In this regard, other 2D materials, including black phosphorus (BP), transition metal dichalcogenides (TMDCs), 2D Te nanoflakes, and so forth, possess advantage properties, such as tunable bandgap, high carrier mobility, ultra-broadband optical absorption, and response, enable 2D materials to hold great potential for next-generation optoelectronic devices, in particular, mid-infrared (MIR) band, which has attracted much attention due to its intensive applications, such as target acquisition, remote sensing, optical communication, and night vision. Motivated by this, this article will focus on the recent progress of semiconducting 2D materials in MIR optoelectronic devices that present a suitable category of 2D materials for light emission devices, modulators, and photodetectors in the MIR band. The challenges encountered and prospects are summarized at the end. We believe that milestone investigations of 2D materials beyond graphene-based MIR optoelectronic devices will emerge soon, and their positive contribution to the nano device commercialization is highly expected.

## 1. Introduction

Two-dimensional (2D) materials have shown intensive talent since the graphene nano flake was synthesized in 2004 [1]. In sharp contrast to conventional semiconductor matters, optoelectronic nanodevices on account of graphene demonstrate distinguished properties, including fast photoresponse speed, nanoscale integrate degree, ultrahigh room temperature carrier mobility, large modulation band width, and so forth [2]. However, the weak light absorption and zero bandgap strictly restrict its further development in optoelectronic applications. Fortunately, tremendous 2D materials have been intensively investigated, such as transition metal dichalcogenides (TMDCs) [3,4,5,6,7], black phosphorus (BP) [8,9,10], perovskite [11,12], and graphdiyne (GDY) [13,14,15], which possess comparable or superior optical, electronic, and optoelectronic properties to graphene, such as tunable bandgaps, high carrier mobility at room temperature, high optical response, ultra-broadband optical absorption, excellent current saturation, high electrical and thermal conductivity, strong luminescence emission, and strong luminescence emission [5,7,8,15], and is widely employed for electronic and optoelectronic applications [16,17,18], including field effect transistors [19,20,21], photodetectors (PD) [22,23,24], modulators [25,26], sensor [27,28,29], logic circuits [19,30,31], and so forth [32,33,34]. Additionally, in sharp contrast to those electronic and optoelectronic devices based on conventional bulk materials, 2D materials based these devices, in particular, whose working wavelength concentrated in the infrared region [35,36], are highly integrated [37,38,39], operated at room temperature, had higher signal-to-noise ratio, had faster response speed [40], etc.

Compared to visible band, infrared light is more important in both civilian and military applications, such as night vision [41,42,43,44,45,46,47], optical communication [48,49,50,51,52,53,54,55], remote sensing [56,57,58,59,60,61,62], target acquisition [63,64,65,66,67,68,69], and so forth. However, infrared light, in particular, mid-infrared (MIR) light, is less investigated compared to visible light. Fortunately, 2D materials can be employed to fabricate high performance MIR optoelectronic devices owing to their superior properties, in particular suitable bandgap, strong light–matter interaction, high on-chip integrate degree, and so forth [42,50,55]. However, more investigations are needed to be carried out to get a better insight into the transport mechanisms, carrier dynamics, and performance of 2D materials-based MIR optoelectronic devices [70,71,72]. In this regard, a comprehensive and detailed understanding of 2D materials beyond graphene-based MIR optoelectronic devices is necessary for the further development of MIR research and technology. Inspired by this point, we summarized the recent development in the field of 2D materials beyond graphene-based MIR optoelectronic devices. Firstly, we briefly summarized the suitable 2D material candidates for MIR optoelectronic applications. Then, we focused on the some recently demonstrated progress of 2D materials beyond graphene-based MIR optoelectronic devices, including light emission devices, modulators, and photodetectors, as summarized in Figure 1. At the end of this review, a perspective on challenges and future development of these attractive materials is also presented.

## 2. Materials and Methods 2D Material Candidates for MIR Applications

In line with density functional theory (DFT) simulations [73], there are almost 5619 compounds with layered structures, and among these layered materials, 1825 compounds can be easily or potentially synthesized [74]. For MIR applications, whose wavelength range from 8 μm to 15 μm, the material shall possess stable absorption and interaction with MIR electromagnetic wave, which means that the band gap of the 2D material must be smaller than the photon energy of the incident light [75]. Consequently, the available 2D materials are matching for MIR applications. Thus, it is meaningful to seek novel 2D materials with excellent properties in the MIR band, in particular, extraordinary optical absorption, long environmental stability, high responsivity, and other excellent optoelectronic properties. In this section, we will summarize the suitable 2D material candidates for MIR optoelectronic applications, including their electric and optical properties.

### 2.1. BP and Related Materials

Since BP flakes were successfully synthesized, it has shown tremendous effect by virtue of its exceptional talent, such as high carrier mobility, excellent nonlinear optical response, strong in-plane anisotropy [76], and thickness dependent bandgap [76,77,78,79,80]. Among these advanced properties, the tunable bandgap is one of the most interesting properties [81,82]. Additionally, with the introduction of the extra modulation processes, such as quantum confinement, high pressure, chemical doping, and high pressure, employing a new tuning knob and mechanical strain means it can be further modified [83,84,85,86,87,88,89,90,91]. Remarkably, BP’s bandgap could be further extended to zero eV via an applied perpendicular bias field, which expands the optical absorption band of BP in MIR and even terahertz (THz) region [92]. The high room temperature carrier mobility and ON/OFF ratio enable BP to possess large sensitivity, fast response speed, and advanced optical absorption efficiency with low dark current, which are desirable for light detection, in particular, MIR detection. Thus, both the widely tunable band gap and excellent electrical properties enable BP as an MIR promising material with high performance. However, the instability issues of BP flakes, caused by oxygen-, water-, and light-induced oxidation, are more and more serious, which severely hinders its further development in both academic and practical applications. To solve this issue, tremendous efforts have been employed to enhance the environmental stability of BP flakes, such as encapsulation, covalent/noncovalent functionalization, and metal/non-metal modification. Recently, BP based composite materials prepared for MIR applications. In a typical synthesis, BP alloyed with As and C, forming b-AsP and b-PC, whose bandgaps were predicted to be 0.15 and 0.59 eV at an optimized composition concentration of b-AsP and monolayer b-PC, respectively [93,94,95,96]. Furthermore, for b-PC condition, the room temperature carrier mobility and the maximum absorbance spectra can be applicable for MIR photoelectric area. For b-AsP condition, even though the room temperature hole mobility was suppressed after the alloying process, it is high than that of TMDCs. However, its easily oxidized properties seriously hinder the following progress in academic and industrial applications [97,98].

### 2.2. TMDCs

The TMDCs have also triggered intensive investigations in MIR band due to their superior photoelectric properties [99,100]. TMDCs stack in X-M-X order, and feature layered hexagonal configurations [101,102]. Interestingly, as the thickness is thinned from bulk down to monolayer, the indirect bandgap turned into direct bandgap, which ensures a stronger photoluminescence effect than that of its bulk counterparts [103]. Moreover, since the excitonic transitions and Van Hoof’s singularity, the optical absorption and light–matter interaction are apparently increased. An optical absorption larger than 10% of monolayer TMDCs was observed at resonance exciton wavelengths [104]. Similar to BP, the bandgap of TMDCs can be effectively adjusted by introducing external modulation processes, such as electrical field modulation and defect engineering. Consequently, TMDCs are considered as promising candidates for MIR photoelectric applications [105,106]. Additionally, most TMDCs possess stable 1 T phase, which enables TMDCs to possess excellent environment stability, which is much better than that of BP flakes and comparable to that of 2D tellurium (Te).

Apart from the TMDCs mentioned above, recently, some noble TMDCs, including ReSe_2_ [107,108,109,110], PtS_2_ [111,112,113,114,115], PtSe_2_ [21,116], PdS_2_ [117,118], PdSe_2_ [118,119,120,121,122], and SrTiO_3_/MoS_2_ [123,124], have attracted much attention due to their excellent environmental stability and suitable MIR bandgaps. Mid-infrared 2D Photodetector based on bilayer PtSe_2_ [125]. Remarkably, the excellent environmental stability (longer than one year) enables PtSe_2_ to hold great potential for practical application in the industry field. For PdSe_2_ semiconductor condition, the bandgap can be altered from 0 to 1.3 eV, which is comparable to that of BP. Furthermore, by lowering the atmospheric pressure, the conductivity type could be altered from p- to n-type of PdSe_2_ [126]. All this outstanding performance indicates that TMDCs are suitable for high performance MIR optoelectronic applications.

### 2.3. Perovskite

Organic–inorganic hybrid perovskite, CH_3_NH_3_PbX_3_ or inorganic perovskite AMX_3_, as emerging novel 2D materials, have attracted intensive attention for ultra-broad band optical equipment owing to the superior optical and electrical performance, such as tunable optoelectronic property, great carrier mobility, and large fluorescence generating efficiency [127,128,129]. Moreover, perovskite meet the demands of the high performance of MIR nonlinear optical applications, including immense nonlinear refractive index [130,131], extraordinary long-term stability [132,133], and high Kerr nonlinearity [134] in MIR band. All these outstanding findings enable perovskite suitable for MIR optoelectronic applications [135,136,137,138].

### 2.4. GDY

As innovative 2D all-carbon materials, large scale GDY flakes were successfully synthesized on the surface of copper in 2010 with a special sp–sp^2^ hybrid structure; they have been intensively explored in various fields due to their outstanding electronic performance, such as the intrinsic bandgap and great carrier mobility [139,140,141]. According to various simulations, GDY possesses a direct bandgap range, which presents an advantage to the zero bandgap of graphene [142,143,144]. The nonlinear absorption coefficients range from visible to MIR band of GDY and were measured to be larger than 10^−1^ cm GW^−1^ [145], which was stronger than that of acknowledged 2D materials, including graphene (−0.66 cm GW^−1^) [146], BP (−6.17 cm GW^−1^) [147], MoS_2_ (−4.6 cm GW^−1^) [148], and MXene (−0.297 cm GW^−1^) [149], indicating that GDY is suitable for MIR ultrafast photonic applications [150]. Noticeably, thanks to the great flexibility of the GDY structure and excellent environmental stability, it can be employed to fabricate high performance and long-term stability of MIR flexible photodetectors [151,152].

### 2.5. Other Candidates

Since then, the 2D Te nanoflakes were successfully synthesized; its alluring performance includes excellent environmental stability, tunable bandgap (0.35 to 1.265 eV), great room temperature carrier mobility, and large photoresponse demonstrate great potential for either academic or industrial applications [153]. Compared with other frequently used 2D materials, 2D Te nanoflakes have an excellent spiral chain shape, which makes them have stable plane anisotropy and great carrier mobility [154,155,156]. Among these fascinating properties, the wide wavelength response, absorption, and steady light–material interaction means 2D Te holds huge potential for MIR optoelectronic applications, including MIR modulator [157], photodetector [158,159], and MIR imaging [160].

Topological insulators (TIs), in particular, Bismuth telluride (Bi_2_Te_3_) [161,162] and Antimony telluride (Sb_2_Te_3_) [163,164,165], have exhibited perfect advantages for MIR applications to its advanced properties, such as, for Bi_2_Te_3_ condition, spectra absorption in a broadband regime convinced by the gapless of Bi_2_Te_3_, strong carrier mobility, excellent signal-to-noise ratio, high light–matter interaction comparable with graphene, and the 2D electron gas enable collective excitation in MIR band [166,167]. For Sb_2_Te_3_ condition, possessing graphene-like electronic-band structure, the linear absorption was higher than that of Bi_2_Te_3_ and Bi_2_Se_3_, indicating the great potential for MIR saturable absorber application [168,169]. Other materials, such as MXenes, can be applied to photoelectrochemical (PEC) and photoelectric sensors, as well as medical equipment [170,171,172]. In addition, Bi-based materials are also commonly used in optoelectronic devices, photodetectors, and lasers [173,174,175]. Phosphate and antimony-based materials are also often used in modulators and sensors [176,177,178,179,180].

To facilitate a better understanding of the spectral range of some representative materials, a comparison of the operation spectra range of materials is listed in Table 1.

## 3. Light Emission Devices

Recently, the property of 2D layered compounds in disparate photoelectric devices attracted the extensive attention of researchers [181]. Carrier concentration and photoelectric characteristics can be adequately handled by adjusting the voltage attribute to the atomic thickness of 2D materials. The 2D materials with a band gap between 0 and 6 eV, such as insulators, semiconductors, semi-metal, topological insulators, and metals, have good electromagnetic spectral response. The electronic band structure of 2D material with layer number dependence, such as the block and monolayer states of BP material, have direct bandgap widths of ~0.3 eV and ~2 eV, respectively. The van der Waals force is weak among 2D material structures. Various van der Waals heterostructures are prepared to meet different requirements and device applications.

### 3.1. Spontaneous Emission

The tiny van der Waals forces between layers allow researchers to stack 2D materials vertically arbitrarily without lattice mismatch and offer great promise in the development of photoelectron detectors and devices [182,183,184]. Among these materials, the tunable direct band gap property between BP layers is recognized to develop an emission candidate in near-infrared and MIR. Yang et al. [185] conducted a comprehensive study about the BP layer amount correlation fluorescence spectrum, and the fluorescence emission showed noticeable layer number correlation characteristics (Figure 2a). The BP layers that range from 1 to 5 have a peak energy range from 0.8 eV to 1.75 eV. BP exciton binding energy band gap is different between the electronic and optical energy band gaps (Figure 2b). The band gap of body blood pressure can be estimated at 0.295 eV by power law fitting. As we know the photoluminescence intensity varies with the number of layers. Zhang et al. [186], confirmed that the peak intensity of photoluminescence increases significantly by reducing layers, as shown in Figure 2c. These results show the number of layers in the BP layer largely determines the quantum efficiency of intrinsic luminescence. The multilayer BP valence band and conduction band valleys’ energy increases the band maximum, resulting in the reduction in the internal quantum efficiency. Finally, the BP photoluminescence intensity is reduced. Sun et al. [187], reported the spontaneous emissivity (SER) of the quantum emitter was significantly improved near the single-layer or double-layer BP. The maximum improvement element for BP is three times that of graphene due to the high photon density nearby. A three-dimensional (3D) figure of the quantum emitter joined to a single layer (Figure 2d) and a double layer BP (Figure 2e) as well as an X–Z plane is shown. In addition, Zhang et al. [188], developed a MIR laser combining a single crystal nanosheet and a segregated Bragg reflection cavity, as shown in Figure 2f–i. The excitation of the MIR at 3611 nm by near uniform linear polarization shows strong thermal stability at temperatures up to 360 K. Furthermore, the cavity length of BP could be changed with modifying the thickness, and the laser wavelength can be increased from 3425 nm to 4068 nm.

**Table 1 nanomaterials-12-02260-t001:** The spectral range of operation of some representative materials.

2D Materials	Bandgap (eV)	The Spectral Range of Operation (μm)	Ref.
BP	0.3~1.5	0.83~4.13	[185]
b-AsP	0.15	8.27	[94]
b-PC	0.59	2.10	[95]
ReSe_2_	1.1~1.58	0.78~1.12	[110]
PdSe_2_	1.3	0.95	[121]
PtSe_2_	0.3	4.13	[116]
GDY	0.46~1.10	1.12~2.69	[141]
2D Te	0.35~1.265	0.98~3.54	[154]
Bi_2_Te_3_	0.21	5.90	[161]
Sb_2_Te_3_	0.45	2.75	[164]

Albert et al. [189], presented research into the electrical characteristics, electroluminescence (EL), and degradation behavior of laser diodes growing on Te-terminated GaAs. Lasers germinated at Te: GaAs had a similar stacking fault density compared to those grown on Zn-treated GaAs but with higher spontaneous emission and a lower threshold of current density. Some of the characteristics of the laser diodes germinated at Te: GaAs are shown in Figure 3a,b. Vu et al. [190] reported a 5 cm ribbed waveguide made by co-sputtering Te and Er and reactive ion etching in an oxygen environment with a peak internal gain of up to 14 dB. The etching technique is an important step in the realization of tellurite waveguide. In previous work, plasma corrosion telluride was used to physically corrode argon causing a propagation loss of 6 dB/cm [191]. Later, excellent tellurite waveguides prepared by RIE were used to reduce the transmission loss below 0.1 dB/cm [192]. Figure 3c,d show the etching of pure TeO_2_ waveguides and Er doped waveguides using this method. In addition, the lifetime data of erbium-doped tellurite films were measured, as shown in Figure 3e. The minimum and maximum pump power of 1.3% Er/Te are expressed (curves (a) and linear (b)). Te has important implications for physics, chemistry, materials science, and, more recently, nanoscience [193,194,195,196]. Choi et al. [197], first disclosed photoluminescence information about Te crystals, as shown in Figure 3f. The photoluminescence and laser emission of Te block crystals and microcrystals were introduced. The photoluminescence of Te block crystals was observed in the medium wave infrared (MWIR) region with a wavelength of 3.75 μm. As the intensity of light excitation increases or the temperature decreases, the MWIR random maser of Te crystal appears at 3.62 μm, as shown in Figure 3f. In addition, the rod-shaped Te microcrystals have secondary and tertiary harmonic lasers in the MWIR and infrared band, respectively.

Li et al. [198], revealed that the self-assembled 2D Ruddlesden–Popper perovskites (RPP) thin films based on methyl ether (FA) and Nai methylamine (NMA) have good optical gain characteristics. They represent a novel solvable material with low threshold, advancing optical stability and multi-wavelength compatibility. Widely studied 3D perovskites exist as free carriers. However, 2D-RPPS show strong bound electron-hole pairs and form a natural energy level gradient. It can realize the ultra-fast energy transfer process from a high energy band gap quantum well to a low energy band gap quantum well, condensing excitons at a lower band gap for easy emission. Therefore, at room temperature, the prepared 2D-RPP films show high optical gain and ultra-low threshold (<20.0 ± 2 μJ/cm^2^) and stoichiometric compatible ASE wavelength, from visible to near infrared (530–810 nm). The optical gains of 2D-RPP thin film (NMA)_2_(FA)Pb_2_Br_7_ and (NMA)_2_(FA)Pb_2_Br_1_I_6_ were up to 330 cm^−1^ and 316 cm^−1^, respectively. In addition, optical tests showed that 2D-RPP thin film (NMA)_2_(FA) PB_2_BryI_7-y_ exhibited good optical stability, with the duration of illumination exceeding 1.2 × 10^8^ laser pulses. Combined with the high electroluminescence efficiency of 2D-RPP films in light-emitting diodes, those solution-treatable 2D-RPP films are expected to realize electric pump spectrum laser.

The GDY is a new type of 2D material, which has had the appreciation of a large number of researchers since it was discovered in 2010. Due to the unique original band gap framework of GDY, its application prospects in optoelectronic devices are better than graphene. GDY has a certain degree of fluorescence, and for the purpose of exploring the photoluminescence properties of GDY, W. Xiao et al., synthesized a GDY hybrid material (FGDY) [199]. In a typical synthesis, xenon difluoride and GDY are heated to obtain FGDY materials and F atoms are doped into the triple bond of GDY, which makes the photoluminescence of FGDY enhanced. According to the different content of the C–F bond in the substance, FGDY can be divided into FGDY-1, FGDY-2 and FGDY-3. Regarding LEDs applications, 2D materials can be introduced as interlayers in LEDs, including anode, HTL (hole transport layer), HIL (hole injection layer), and EIL (electron transport layer), and the performance of the device can be significantly enhanced due to the improved work function, effective electron blocking, and the increased hole injection from anode into the organic layer, which enable more holes and electrons recombining in emission layers. Moreover, the electron injection can be enhanced as well when 2D materials are employed as dopant in EIL. The proposed photoluminescence of FGDY has application prospects for GDY in the field of LEDs or sensors. After that, in order to study the application of GDY in MIR optics.

Recently, 2D TMDCs, including MoSe_2_, MoS_2_ and WSe_2_, due to their sharp linewidth emissions on the lower energy side of the delocalized exciton emission, neutral excitons trapped at anisotropic confining potentials from defects are suitable for single photon emitter applications [200]. Meanwhile, combined with the plasmonic effect and the strain engineering, the single photon emitter performance can be further enhanced. The single photon emission was verified to originate from spatially localized regions of the TMDCs samples. These results suggest that single photon infrared emitters can be realized in 2D materials and have promising applications in quantum devices.

To facilitate a better understanding of the 2D materials for LEDs operation, potential 2D material candidates are listed in Table 2.

### 3.2. Laser

In recent years, while the graphene industry is booming, another new single-element 2D atomic crystal material, BP, has been discovered. Similar to graphene, BP has many excellent properties, so it is called a “dream material” compared to graphene. The research and application of BP has just begun, and its nonlinear optical properties have been confirmed by many domestic and foreign units and applied to the production of ultra-fast lasers. In the foreseeable future, it will be the “second graphene”. Mu et al. [202] found that BP has the characteristics of wide-band saturable light absorption and the wavelength range can cover the visible light to the MIR band. In laser field, components possess high saturable absorption, which is suitable for constructing ultrashort pulses [203,204,205]. The discovery of BP provides a possibility for the development of mid-infrared ultra-fast devices. The most important feature of BP is that it has a direct band gap varying with the layer number, which solves the problem of graphene.

Li et al. [205], prepared BP 2D materials using an economical and effective liquid phase exfoliation (LPE) method. BP saturated absorber (SA) was applied to two novel medium infrared rare earth fluoride fiber lasers. At the same time, its saturation absorption was studied in the 2 μm spectrum. The Q switch and mode locking were obtained by specially designing the cavity structures of two fluoride fiber lasers based on the same BP, which firstly extended the feasible operating wavelength to 3 μm. In addition, for the generated pulses, the functional wavelength at about 2970 nm is the maximum wavelength of the fiber laser using BP SA. In Figure 4, the continuous wave (CW) laser can be generated by increasing the transmitter pump power to 302.6 mW. An increase in pump power to 489.3 produced a repetition frequency of 12.43 kHz, as can be seen from Figure 4a. It is manifested in Figure 4b that the laser remains in a stable operation of Q-switching and there is change at the maximum transmitter pump power of 2.99 W. Figure 4c displays the test spectrum of pulse, and the spectrum of CW laser is also observed. The central wavelength is red shifted to 2972.8 nm, which is due to the reduction in the initial Stark manifold at the ^5^I_6_ level as the reduction in the loss in the cavity. According to Figure 4d, the radio frequency (RF) spectrum was also measured at a 45 kHz sweep span and a 100 Hz resolution bandwidth at 2.99 W of transmitted pump power. In this wavelength range, the signal-to-noise ratio of 37.7 dB is at the average of passive Q-switched fiber lasers [206,207,208,209,210]. Figure 4e reveals the functional relationship between tested repetition frequency and pulse duration and starting pump power. Figure 4f shows the tested output power and the theoretical monopulse energy. Huang et al. [211], developed a compact economic surface-emitting laser in the 3–5 μm MIR range. The manufacturing process of the schematic diagram and the BP surface-emitting laser of the microscopic image are shown in Figure 5a–c. MIR surface-emitting laser with BP based was demonstrated firstly at room temperature, using BP as the gain material inserted in SiO_2_/Si_3_N_4_ to create an open microcavity on silicon. By significantly increasing the luminous efficiency of BP layers and solving the common issue of processing BP and other 2D materials into gain media. Based on the special design of open cavity, an optically pumped laser of about 3765 nm is achieved. This research has significant implications for gas detection, non-invasive medical detection, and infrared band projection applications. Similarly, Qin et al. [212], successfully made the medium infrared saturable absorption mirror. A 2.8 μm Er: ZBLAN fiber laser at 613 mW maximum average output power 24 MHz repetition frequency, and 42 ps pulse duration of were demonstrated using a BP saturable absorption mirror. A BP based mode-locked laser at 2.8 μm has been proposed for the first time, and the feasibility of using BP thin slices as a new 2D material in MIR ultra-fast photonics has been proved.

As one of 2D materials, TMDCs can be applied to optoelectronic equipment. Because of the special electrical properties of TMDCs, it has greater application prospects in the realm of optoelectronics. For example, to explore the potential of MoS_2_ in lasers, Tian et al. placed a 2D multilayer MoS_2_ on a mirror for experiments [182]. The nonlinear absorption curve of the experiment is shown in Figure 6a. Among them, MoS_2_ is prepared by LPE, and the material of the mirror is gold. In this experiment, the phenomenon of saturated absorption is observed. Multilayer MoS_2_ with a thickness of 2 μm is used as the model-locking function of Tm^3+^-doped fiber lasers. Figure 6b presents that the pump power is 430 mW. The output power of the laser is positively related to the pump power. The output of the laser can reach 150 mW at the pump power reach to 700 mW. Figure 6c shows the laser pulse sequence diagram of the laser. Therefore, MoS_2_ is a potential material for broadband mode-locked laser. Wang et al., studied the saturation absorption of MoS_2_ and applied it to 2.8 μm fiber laser [213]. In the experiment, multilayer MoS_2_ films were obtained by ultrasonic treatment. After that, MoS_2_ was painted on the Au mirror surface. The saturated absorption performance of the MoS_2_ film was measured by reflection method and it was 2.8 μm, and the results of the saturated absorption measurement data are shown in Figure 6d. This shows that MoS_2_ can be used as a Q-switch for Er^3+^-doped fiber lasers, and the researchers have conducted experiments on this. The output power is positively related to the pump power, and the experimental data are demonstrated in Figure 6e. The output power can reach 140 mW, and the radio frequency spectrum measurement as the maximum output power is demonstrated in Figure 6f, and it shows that MoS_2_ can work stably as a saturable absorber with a 2.8 μm fiber laser. In order to compare the property of several TMDCs used in saturable absorbers, Chen et al., compared MoS_2_, MoSe_2_, WS_2_, and WSe_2_ [214]. These four substances are used in saturable absorbers with the same structure, and these materials are combined with polyvinyl alcohol materials (PVA). Figure 6g shows the preparation process of the TMDCs–PVA materials, and Figure 6h shows the images of different TMDCs–PVA materials. The structure of a fiber laser with TMDCs–PVA as a saturable absorber is shown in Figure 6i. The outcome demonstrates that the results show that MoS_2_ has the best modulation depth, and the pulse sequence of WS_2_ is the most stable. This work proves the application potential of TMDCs as saturable absorbers for fiber lasers.

TIs is a kind of insulating material, and its interior is insulating, and its surface is conductive, ascribed to the existence of its special quantum states. Band gap of 2D TIs is relatively narrow, so TIs have better optical absorption in the MIR range, which allows TIs to be used as saturable absorbers with better performance. Tang et al. [215], used Bi_2_Te_3_ in the saturated absorption part of the Er^3+^ fiber laser, and the performance of the laser was measured. In the experiment, Bi2Te3nanosheet was used as the center material, and polymethyl methacrylate (PMMA) was used as the outer layer material. In this way, a PMMA sandwiched TI: Bi_2_Te_3_ material is obtained. Figure 7a demonstrates the transmission spectrum of Bi_2_Te_3_ material in the range of 2000–3000 nm. Figure 7b exhibits the trend of the output and pulse power with the input pump power. Once the pump power increases to 5.9 W, the output power can reach 856 mW. These data are thirteen times more than that of graphene as the saturable absorber of the traditional device. Figure 7c demonstrates the output spectrum of PMMA sandwiched TI: Bi_2_Te_3_ as a saturable absorber when the pump power is 5.9 W. At the same time, the SNR of fiber laser is low, which can indicate that Bi_2_Te_3_ has better working stability for lasers. These findings confirm the application potential of Bi_2_Te_3_ in the area of 2.8 μm lasers. In addition, in the field of 3.0 μm laser, Li et al., used Bi_2_Te_3_ as the saturable absorber of the laser [210]. In this experiment, calcium fluoride was used as the substrate, and the Bi_2_Te_3_ nanosheet was prepared by the intercalation/stripping method. After that Bi_2_Te_3_ nanosheet was mixed with ethanol solution, and the solution was coated on calcium fluoride. This material is used as a saturable absorber for the laser. The structure of the experiment platform is shown in Figure 7d, where LD represents a laser diode and PBS is a beam splitter. The repetition frequency and pulse duration varied with pump power is obtained in Figure 7e, and the relationship of the average power of the output, pulse energy, and pump power are demonstrated in Figure 7f. In the input pump power range 3.0–3.5 W, the output power is around 327.38 mW, and the maximum pulse energy is 3.99 μJ. This work demonstrates the application capability of Bi_2_Te_3_ as a saturable absorber in the 2979.9 nm band. In order to reduce the optical components required for q-switched lasers, Li et al., applied Bi_2_Se_3_ to small fiber lasers [216]. The few-layer Bi_2_Se_3_ manufactured by exfoliation is used as a saturable absorber. The structure of the 3 μm laser is shown in Figure 7g. The two illustrations are the images of the fiber end mirror M1 and M2. In the experiment, a pump laser of 1153 nm was served as the light source. The MIR pulse appeared when the pump power was 48 mW, and the critical value of 70 mW could obtain a stable pulse. When the pump power reaches 194 mW, the output spectrum curve is demonstrated in Figure 7h. The relationship of pulse time, repetition rate and pump power are exhibited in Figure 7i. Since the pump power is increased to 230 mW, pulse duration will reduce to 1.5 μs, and the pulse repetition rate increase by 55.1 KHz. The method proposed in this work can effectively reduce the size fiber laser.

Te is the attractive material used to detect and monitor the CO_2_ laser radiation [217,218]. Ribakovset et al. [219], used a sub nanosecond detector and beam monitor for pulsed CO_2_ laser radiation using photonic resistance and optical rectification effects in Te. Although the photon dragging effect in Te is strong enough to act as an electromotive force source for high power pulsed carbon dioxide laser radiation. However, mechanisms characterized by third-order tensor coefficients lead to greater electromotive force. In recent years, Bi_2_Te_3_, Sb_2_Te_3,_ and other fiber lasers used for Q tuning or mode-locking have been widely reported in 1.2 μm [220,221,222,223,224,225]. Similar to graphene, perovskite materials have ultra-wide saturation absorption bands and are used in ultrashort pulse lasers due to their high modulation depth and damage threshold [226,227,228]. Li et al. [210], first reported that the 3 μm band is based on Bi_2_Te_3_ passive Q-switched fiber laser, as shown in Figure 8. Bi_2_Te_3_ nanosheets were prepared by hydrothermal method and deposited on the CaF_2_ substrate to form free space SA. The modulation depth (MD) and saturation intensity were 51.3% and 2.12 mW·cm^−2^, respectively. Two 1150 nm laser diodes were coupled to a 5.2 m Ho^3+^: ZBLAN gain fiber through a polarization divider to obtain a pulse output at central wavelength of 2979.9 nm. The maximum repetition frequency and the minimum pulse width were 81.96 kHz and 1.37 μs, respectively. The average output power was 327.38 mW with monopulse energy was 3.99 μJ in maximum. The Q pulse width and repetition frequency can be narrowed by selecting Bi_2_Te_3_ with a certain modulation depth and an appropriate cavity length. In the experiment, due to the large modulation depth and unsaturated loss, the pulse energy in the cavity is low, and it is difficult to achieve the continuous mode-locking condition, so the mode-locking state does not appear in the experiment. Sotor et al. [225], proposed the effective mode-locking erbium-doped fiber laser based on a special saturable absorbent material, antimony telluride. Sb_2_Te_3_ layers were obtained by mechanical spalling method 15–17 commonly used for graphene. A view of the surface of antimony telluride under an atomic force microscope is shown in Figure 8d. The core surface is almost completely covered by the Sb_2_Te_3_ layer. The composition of the sediments was confirmed by the energy spectrum obtained by Hitachi SU 6600 scanning electron microscope. It is then transferred to the fiber connector tip by adjusting the appropriate Sb_2_Te_3_ layer thickness. The all-fiber laser can generate solitons with a half-peak width of 1.8 nm (Figure 8e,f).

High power medium infrared fiber laser has important applications in frontier scientific research, biomedical, environmental monitoring, infrared remote sensing, infrared imaging, atmospheric communication, photoelectric countermeasures, and many other fields [229,230,231,232,233,234,235]. The medium infrared fiber laser material is the core component of the medium infrared fiber laser. The medium infrared fiber laser can be built by using the rare earth ion-doped medium infrared fiber as the gain medium, and the wide band medium infrared laser can be output by further utilizing the nonlinear optical effect in the medium infrared fiber. After decades of development, only zirconium fluoride glass (ZrF_4_-BaF_2_-Laf_3_-AIF_3_-NaF, ZBLAN) can be used as the high-power medium infrared fiber laser material at the average power of 10 W and a working central wavelength greater than 2.5 μm laser output. However, the researchers later found that the end face of ZBLAN optical fiber is easy to react with water in the air due to its poor anti-tidal ability. This will result in optical fiber end face damage, which will affect the practical application of ZBLAN fiber based high power MIR laser. In addition, the conversion temperature of ZBLAN glass is only 252 °C, and its thermo-mechanical quality factor is low, which affects the increase in the power of the medium-infrared ZBLAN fiber laser. To solve the listed problems, Yao et al. [236], screened a new generation of high-power medium infrared fiber laser material fluorotellurate glass through component optimization. The main group is divided into TeO_2_-BaF_2_-Y_2_O_3_ (TBY). TBY glass not only has a wide optical transmittance window (covering 0.4~6 μm band), but also has strong moisture resistance, high glass transition temperature (~424 °C), and high thermal mechanical quality factor. The glass transition temperature is 172 °C higher than ZBLAN glass, and the thermal mechanical quality factor is 1.5 times higher than ZBLAN glass. Further, the screening results showed that the low refractive index glass AIF_3_-BaF_2_-CaF_2_-YF_3_-SrF_2_-MgF_2_-TeO_2_ (ABCYSMT) was well matched with the thermal properties of TBY glass. TBY and ABCYSMT glass were selected as fiber core and cladding materials respectively, and high-quality fiber prefabrication rod was prepared by inhalation method. Low-loss fluorotellurite glass fibers (Figure 8g,h) with dispersion, nonlinearity, and numerical aperture adjustable in a wide range were prepared by optimizing the wire drawing process. For the purpose of verifying the performance of the high-power medium-infrared laser based on this kind of fiber, the all-fiber medium-infrared supercontinuum laser was designed and built, in which the pump source was 2 m high-power fs fiber laser and the nonlinear medium was fluorotellurate glass fiber with core diameter of 6.8 m. When the pump power was 15.9 W, an all-fiber super-continuous spectrum laser output at the average power of 10 W and a working wavelength of 0.95~3.9 m was obtained, which could operate stably for a long time. The experimental results of high power 2 m laser transmission show that the use of fluorotellurate glass fiber as a nonlinear medium is expected to achieve tens or even hundreds of watts of magnitude, long time stable operation of the medium infrared supercontinuum laser, which provides an opportunity for the high-power all-fiber MIR supercontinuum laser.

To facilitate a better understanding of the laser operation based on 2D materials, a comparison of the pulse duration of the several lasers and their threshold is listed in Table 3.

## 4. Modulator

Light modulation is a crucial step in optics. It is widely used in optical communication, pollution monitoring, biological research, medical, and safety applications. In an era of continuous and rapid development of information and internet applications, such as streaming media and cloud computing, and connections between network data, including those between traditional data networks, as well as internal chips, are growing exponentially. The main cable connection (such as copper cable) has different performance limitation in all aspects, and there is an urgent need for better connectivity methods. Therefore, the research of optical modulator is more and more important. The need for high performance, compact, low cost, efficient, fast, and high bandwidth optical modulators [237] is becoming more and more urgent, especially in light transmission [238], lasers [239], biology [240], and medical treatment [241].

2D layered materials possessing modulation impact is one of the hot topics in the last decade. Lots of reports demonstrated that 2D modulators can generated varying modulation mechanisms (such as all-optical, electro-optical, and thermotical modulation), making the 2D layered materials highly competitive. For example, graphene optical modulators have extremely wide operating bandwidth (ranging from visible light to microwave region [242,243]). The 2D materials, such as graphene and their isomorphic saturated absorbers, have been used to generate ultra-high-speed pulses in a variety of lasers and have shown excellent performance. There is growing interest in the commercialization of various laser applications, particularly in ultra-fast laser sources with high repetition rates for optical interconnections [244,245]. Nonlinear wavelength modulators and electro-optical amplitude modulators based on 2D materials have also been used for high-speed data transmission, and high-speed optical modulation technology in optical interconnection has been further improved.

### 4.1. All Optical Approach

All-optical modulation techniques for 2D layered materials have been developed rapidly and signal processing in the photonic domain has become more accurate. Optical modulation can be performed in optical fiber systems (such as silicon waveguides) and enables ultra-fast, low-loss, and wide-band optical signal. All-optical modulator-based 2D materials, include saturable absorber [246,247,248], wavelength converter [249], optical limiter [250], and polarization controller [251], etc. These devices take advantage of 2D materials’ nonlinear optical properties (primarily third-order susceptibility), such as wide band, fast response, small size, compact, and integrated full optical operating characteristics. At present, there is a growing demand for ultrafast lasers. The 2D material-based saturable absorber is one of the commonly used optical devices. The 2D material-based non-originating amplitude modulator can generate ultra-fast pulses [252,253]. The band structure of graphene ensures that the electron-hole excitation is present in any incident photon. In the ultra-fast time scale, the interaction between carriers and ultrafast optical pulses produces unbalanced carrier groups in the baseband and the guide bands [254]. This result ensures Pauli-blocking broadband and ultra-fast saturation absorption. However, when the wavelength is smaller than the NIR region, the saturated energy density is relatively large, which prevents the application of graphene at the end of the spectrum. Unlike graphene, TMDs [255,256] and BP have a band gap for resonant light absorption. TMDs [257] usually have resonance absorption in visible light, while BPus [258] shows resonance absorption in near-infrared and MIR regions. Thus, within this wavelength range, graphene saturable absorbents have suitable substitutes.

TMDCs materials, such as MoS_2_, WS_2_ and MoTe_2_, can be used in the all-optical modulation method of the modulator. For example, Ahmad et al., applied MoS_2_ film to the polarization modulation system [259]. In this experiment, the refractive index and thickness of MoS_2_ can be adjusted by changes in temperature. The Raman spectroscopic characterization of the prepared MoS_2_ film is demonstrated in Figure 9a. The azimuth and ellipticity will change with the thickness of the MoS_2_ ¬film when the pump power increases, as shown in Figure 9b. As the pump power changes, the azimuth and ellipticity change values are relatively small, which shows that the modulator can work stably. Figure 9c demonstrates the change curve of output power and ellipticity in the range of pump power from 0–550 mW. This work effectively reduces the design complexity of the laser cavity, and it can be applied to all-optical system. Yang et al., integrated the 2D TMDCs material WS_2_ and silicon nitride as the all-optical modular of the pump light source and conducted experiments [260]. Compared with BP and MoTe_2_, the chemical properties of the WS_2_ material are more stable, and compared with MoS_2_, its luminous efficiency is higher. In this experiment, the oxidized silicon plate is used as the substrate, and WS_2_ and silicon nitride are integrated on the substrate. This modulator material is named Si_3_N_4_-WS_2_-Al_2_O_3_, and Figure 9d shows the microscopic image of Si_3_N_4_-WS_2_-Al_2_O_3_ illuminated by 532 nm light source. The researchers chose a 532 nm laser as the pumping source, and the light across the band pass filter and the acousto-optic modular to the beam splitter. The modulation signal is sent out by a 640 nm laser and merges with 532 nm light through the beam splitter. After that, the light pass through the modulator and reaches different receivers through the beam splitter, and the structure of the test platform is demonstrated in Figure 9e. Figure 9f demonstrates the change in the emission spectrum when the modulator material is changed. The volume of the modulator is small, and the optical structure can be changed to improve the modulation effect, so its application potential in the field of integration is better. In addition, in order to study the application potential of TMDCs in THz modulators, Qiao et al., combined MoTe_2_ with silicon as an all-optical THz modulator [261]. In the experiment, the co-solvent method and the liquid phase peeling method were used to prepare MoTe_2_ flakes, and the MoTe_2_ solution was coated on the Si substrate. After heating and baking the silicon substrate, the MoTe_2_/Si heterostructure is obtained. The comparison image of silicon substrate and MoTe_2_/Si is shown in Figure 9g. The THz signal with the different power of 1064 nm laser irradiation is shown in Figure 9h. When MoTe_2_/Si is used as a modulator under the power of less than 300 mW, the modulation depth can reach 99.9%, as shown in Figure 9i. In addition, the MoTe_2_/Si heterostructure modulator can perform normal modulation work within 0.3–2.0 THz. The manufacturing method of this all-optical THz modular is simple and has high practicability.

Although BP is widely used as a promising nanomaterial, the practical application of BP with large area and few uniform layers is severely limited by its inherent defects and irreversible oxidation in the synthesis process. Aiming at the above problems, Zheng et al. [262], successfully prepared large area and few layers of BP by using electrochemical cathode stripping method combined with centrifugal technology. The composite structure of BP-micro-nano fiber is constructed and applied to all optical signal processing successfully. The team successfully prepared large area and few layers (mainly 4 layers) of BP using an electrochemical cathode stripping method combined with centrifugal technology. Then, a small amount of BP material was deposited in the pull-cone region of the micro-nano fiber. The taper fiber is used as an optical waveguide to realize the stable transmission of light in the micro-nano optical fiber. With the interaction of evanescent field on the surface of micro-nano fiber and low layer BP material, the BP carrier will have interred band transition under high power laser pumping. During the relaxation time of the carrier, the system no longer absorbs the other light, thus realizing the saturated absorption characteristic of BP, as shown in Figure 10a–c. Therefore, for the first time, an all-optical threshold device that can suppress noise and improve the signal-to-noise ratio of optical pulses is realized. The experimental results show that the SNR increases from 3.54 to 17.5, as shown in Figure 10d. BP has poor environmental stability and is easy to oxidize in air and water, so it usually needs more complex protection measures. By means of synchronous fluorination electrochemical dissection, Wang et al. [263], successfully prepared a 2D material with a low layer of BP with high stability, as shown in Figure 10e. An all-optical modulator consisting of a Mach–Zehnder interferometer was successfully constructed by using the photothermal effect of BP, and the phase modulation and intensity modulation characteristics from an all-optical modulator were systematically studied. Finally, a list of ASCII code signals is loaded from the pump light to the signal light in the all-optical domain, as shown Figure 10f. The prepared BP layer is between three and eight, and it has obvious photothermal effect under the pump light irradiation. The phase shift and conversion efficiency of all-optical modulator are up to 8 and 0.029 π/mW respectively, and an intensity modulation depth of up to 17 dB. In addition, the rise and fall time constants were 2.5 ms/2.1 ms, respectively, and the system still had good performance after two weeks in the laboratory environment. Similarly, Zhang et al. [264], experimentally constructed a BP wide-band light modulator and used BP optical modulator as a saturated absorber to achieve passive modulated bulk lasers of 639 nm (red), 1.06 m (NIR), and 2.1 m (MIR), as shown in Figure 10g–i. In addition, by optimizing sampling and designing the laser cavity, the performance of pulsed lasers can be improved. This opens a new path for universal optical modulators that will facilitate BP’s further application beyond existing optoelectronic devices.

The 2D Te material is a kind of graphene, composed of a single atom layered structure material. In the past two years, it has been predicted and successfully prepared. With its excellent photoelectric properties, it has rapidly attracted extensive attention and research in the academic circle. At present, the research and application of 2D Te material properties are still in its infancy, although it provides a wide development future in catalytic, sensing, optical components, and other aspects. Firstly, the nonlinear optical parameters of the 2D Te material were measured, and it was found that the 2D Te material had a very strong nonlinear optical response. By using this property, Wu et al., further studied the important application of 2D Te in nonlinear optical devices. They cleverly combined this 2D Te material with a strong nonlinear effect with another 2D material, such as SnS_2_, with the characteristics of trans-saturation. After combining these 2D materials, they realized the non-reciprocal propagation of light, which opened up a new idea for the application of photonic diodes (Figure 11a–c). Secondly, they also explored the important application of 2D Te material in all-optical modulation/switch and found that 2D Te could be used as the carrier of information transformation to realize the modulation of the pump light to detect light, thus realizing the conversion between “on” and “off” in the optical switch (Figure 11d–f). The 2D Te has wide band response, absorption, strong light and material interaction effects, and contact environmental stability, which ensures that 2D Te can be used to prepare high-performance light modulators. Guo et al. [265], introduced a saturated absorber based on 2D Te (Figure 11f), with the frequency and peak-background ratio of 15.45 mhz and 53 dB, respectively (Figure 11g). In addition, the pulse train produced has no obvious fluctuation, demonstrating its excellent stability during the laser mode-locking process.

Halide perovskite exhibits outstanding photoelectric characteristics and is used in photoelectric detector and luminescent display field [267,268]. Most of the above research focuses on single crystal and polycrystalline thin film materials of perovskite. With further research, researchers began to pay attention to the luminescence characteristics of perovskite nanomaterials. In 2014, Schmidt et al. [269], from the University of Valencia in Spain reported the synthesis and fluorescence enhancement of perovskite nanoparticles. The research on luminescence application of perovskite nanomaterials has been developed rapidly. It has been possible to accurately control the morphology from zero win Amite crystal and one win Amite wire to a 2D nanometer sheet [270], and the understanding of its size and dimensionally dependent optical properties has also been further developed. On the other hand, the luminescence and device application technologies of perovskite nanomaterials have also been developed rapidly, among which the photoluminescence and electroluminescence technologies based on perovskite quantum dots have attracted the most attention. Metal halide perovskite materials have excellent photoelectric properties. However, the luminous efficiency of the bulk perovskite materials is very low, which cannot meet the application requirements of electroluminescence and laser. Compared with block materials, perovskite nanomaterials have higher quantum yield, narrow half-peak width, and wide spectrum control range and have shown great potential in future display technology applications. Compared with ordinary zero win Amite crystals, nanocrystals provide 2D exciton luminescence characteristics. In addition to high quantum yield, narrow half-peak width, and polarization luminescence characteristics [271], the fluorescence spectra of 2D perovskite materials can be adjusted by layer number and halogen type [272]. Figure 12a,b shows the emission spectra of 2D perovskite nanomaterials of different halogen elements [272,273]. As nanosheets thickness is more than three, photoluminescence quantum yield (PLQY) > 70%. PLQY for n = 2 and n = 3 is only about 5~20%. When the number of nanosheets n = 1, PLQY < 1%. This could be because of the weak dielectric shielding effect, which makes it difficult for polarized photons to scatter at the defect to be shielded. On the other hand, when the number of layers n = 1, 2, the blue light emission of br-based perovskite can be realized by using the 2D perovskite quantum confinement effect, which indicates that the quantum confinement effect of 2D perovskite is more obvious than that of quantum dots. The 2D perovskite surrounded by organic media has a smaller dielectric constant, making it more difficult to screen out coulombic interactions between electron holes. As a result, the binding energy between excitons will far exceed that of bulk materials. Although the exciton binding energy of block materials is generally 5~60 meV, theory predicts that the exciton binding energy of 2D lamellar perovskite materials may reach 200~500 meV [272]. In terms of photoluminescence, Zhang et al. [274], invented ligand-assisted reprecipitation technology for perovskite quantum dots and in-situ preparation technology for optical film of perovskite quantum dots (Figure 12c,d). The colloidal CH_3_NH_3_PbX_3_ (X = Br, I, Cl) quantum dots with bright luminescence and adjustable color were achieved with ligand-assisted reprecipitation. The absolute quantum yield was as high as 70% at room temperature with low excitation flux. A comprehensive composition and surface characterization was performed to explain the photoluminescence intensification in these quantum dots, and the photoluminescence spectra associated with time and temperature were determined. Further, a wide-color domain white light emitting diode based on green CH_3_NH_3_PbBr_3_ quantum dots and red K_2_SiF_6_: Mn^4+^ as color converters has been shown to enhance the color quality of the display technology (Figure 12e–g). Li et al. [275] made important progress in the covering of perovskite quantum dots, which promoted the progress of chip-encapsulated LED backlight. QD mixed with silica/alumina (QDS-SAM) was successfully prepared by a simple sol–gel reaction of Al–SI single precursor with CsPbBr_3_ QD mixed in toluene solution (Figure 12h). The obtained transparent aggregates exhibit a 90% PLQY and a significant degree of optical stability under intense blue light illumination for 300 h. Qds-SAM in the form of CsPbBr_3_ powder can be easily blended into the resin and used as a color conversion layer and cured on blue LEDs. The material shows good luminous efficiency and narrow emission of 80 lm W^−1^ with a half-peak full width (FWHM) of 25 nm (Figure 12i).

### 4.2. Electro-Optic Approach

Many compact systems that use MIR technology still face compatibility problems when integrated with conventional electronics. Because of its various uses in photonic circuits, BP has gained attention for overcoming these challenges. Lin et al. [276], designed an electro-optical modulator based on BP in the MIR band (Figure 13a,b). The BP is coated on the silicon waveguide, and the two electrodes contact the silicon waveguide, respectively, so as to change the electrical interference of BP. The BP is coated with Al_2_O_3_ to prevent it from being oxidized. The silicon waveguide thickness is 50 nm, and the BP thickness is 0.7 nm. Under the action of bias voltage, the absorption of BP will be red shifted, blue shifted or bidirectional shifted. The absorption of light is related to the thickness, doping, and working band of BPus. However, MIR electro-optical modulation relies on narrow-band compound semiconductors and is difficult to integrate with uneconomical silicon photons. BP has been shown to be a promising 2D material for medium infrared light detection. Peng et al. [277] tested the electro-optical modulation of MIR absorption in several layers of BP, as shown in Figure 13c. The experimental results show that the quantum-limited Franz–Keldysh effect is the key mechanism of electro-optical modulation in the available doping range. The spectral analysis of samples with different thickness shows that certain sub-bands have strong interlayer dependence on inter-band transitions. Figure 13d,e are optical microscope images and atomic force microscope images of the BP modulator. In order to effectively adjust the band gap, Deng et al. [278], demonstrated the unique band gap tuning characteristics caused by strong interlayer electron state coupling in the inherent BPus in Figure 13f. In addition, a 10 nm thick BP was used to optimize the band gap from 300 to 50 mev, and a 1.1 v nm^−1^ dielectric displacement field was used (Figure 13g). The thickness of BP can be measured by atomic force microscope (AFM), as shown in Figure 13h. The dynamic tuning of the bandgap extends the working wavelength range of tunable BP sub-devices and provides a direction for the research of tunable topological insulators and semi-metals.

For the purpose of reducing the volume of optoelectronic devices, Li et al., combined MoS_2_ and Au to obtain an electro-optical modulator (Au/MoS_2_) [279]. MoS_2_ is used as the substrate, and the dish-shaped Au material is installed on the MoS_2_, as shown in Figure 14a. Among them, MoS_2_ is manufactured by vapor deposition method, meanwhile the Au dish is photoetched on MoS_2_. Figure 14b shows the scattering spectra of the Au dish and Au/MoS_2_ and the absorption spectra of MoS_2_ films. Fano resonance is generated by excitons and plasma as MoS_2_ and Au dish. The change of the applied voltage can tune the Fano resonance phenomenon in the Au/ MoS_2_ modulator, as shown in Figure 14c. A voltage with a period of 1 ms is applied to the Au/MoS_2_ modulator, and the switching time of the modulator is less than 200 ms. This design can realize nano-level optically sensitive display devices. WSe_2_ materials can be used in electrical switches and photonic devices. Seyler et al., proposed a method to apply WSe_2_ to electro-optic modulators [280]. A single layer of WSe_2_ material is processed by photolithography on the nitrogen-doped silicon oxide layer, as shown in Figure 14d. The exciton charging effect in the single-layer WSe_2_ is used to tune the oscillator strength. In the experiment, the second-harmonic-generation (SHG) phenomenon was observed at an excitation voltage of 1.66 eV, as shown in Figure 14e. Figure 14f shows the SHG spectra as different voltages. This work confirmed the application potential of TMDCs in nonlinear optical devices. Similarly, Yu et al., used double-layer WSe_2_ to obtain SHG, which is called charge-induced SHG (CHISHG) [281]. The double-layer WSe_2_ used in the experiment was made by a mechanical peeling method, and WSe_2_ was used to make a FET for SHG observation, as shown in Figure 14g. Figure 14h is a diagram of the relationship between excitation energy and CHISHG intensity, it can be seen that the two are secondarily related. In addition, the SHG signal intensity of single-layer WSe_2_ and double-layer WSe_2_ in the same experimental conditions are compared, as shown in Figure 14i. The signal strength of the former is about 1000 times than that of the latter as a gate voltage of −40 V. This work expands the potential applications of TMDCs in electronic devices.

In recent years, the fluorescence quantum yield of 2D perovskite nanosheets can exceed 80%. The high fluorescence quantum yield and the flexibility of the structure mean 2D perovskite materials have a broad prospect in light-emitting devices. In 2016, Kumar et al. [282], of the Swiss Federal Institute of Technology, prepared 2D perovskite materials with the number of layers of n = 7~10, n = 5, n = 3, n = 1 by precisely adjusting the number of layers. Simultaneously, 2D perovskite electroluminescent devices with different layers were prepared. Especially for devices with a layer number of n ≤ 3, blue light emission can be realized, which fills the blank of 3D perovskite in blue light electroluminescent devices. In addition to adjusting the number of layers to regulate the wavelength of light, the regulation components can achieve a wider range of light wavelength regulation. Congreve et al. [283] prepared a series of electroluminescent devices by adjusting the type of halogen ions, and the emission wavelength can be adjusted between 440 and 650 nm. Recently, Kumar et al. [284], of the Swiss Federal Institute of Technology, prepared a high-purity green electroluminescent device based on perovskite nanocrystals. Its gamut can reach 97% of the Rec.2020 gamut, the current efficiency is 13.02 cd/A, and large area and flexible devices are realized (Figure 15a,b). Hu et al. [285] deposited CsPbBr_3_ nanometer tablets on the ITO surface by chemical vapor deposition and patterned the electrodes on the ITO surface, as shown in Figure 15c. The CsPbBr_3_ nanometer sheet single particle electroluminescent device was realized, with an opening voltage of 3 V and an external quantum efficiency of 0.2%. Figure 15d shows the diffusion of the transmitting point at a high bias voltage. The high-quality optical cavity inside the nanoplate results in strong waveguide emission. Wu et al. [286], introduced a novel perovskite sample, K_3_B_6_O_10_Cl, which showed a rapid second harmonic response than that of KH_2_PO_4_ (KDP). The perovskite framework represented by ABX3 is formed by the vertex connection between the hexaborate [286] group at site A and the octahedron group centered on BX_3_Cl (Figure 15e,f).

### 4.3. Other Approaches

Modulators that use magneto-optical effects for optical modulation have received less attention than all-optical or electro-optical modulators. Although the operation of magneto-optical modulation is relatively simple, the magneto-optical modulator has a unique non-reciprocity that other modulators cannot achieve. Magneto-optical Faraday rotators and Kerr rotators [287] can operate in the far infrared, terahertz, and microwave ranges. In 2D materials, the properties are different because of the number of layers. In contrast to the single-layer/three-layer CrI_3_, the double-layer CrI_3_ exhibits antiferromagnetic behavior due to the lack of off-plane and in-plane magnetization [288]. This antiferromagnetic behavior can also be tuned to ferromagnetic by applying a gate voltage. Huang et al. [289] demonstrated the magneto-optic Kerr effect (MOKE) microscopic detection of double layers with magnetic electrostatic door control in CrI_3_, as shown in Figure 16a,b. The voltage control switch between antiferromagnetic and ferromagnetic states is realized at the fixed magnetic field near the sub magnetic transition. The stratified antiferromagnetic state of time reversal pair is verified under zero magnetic fields. The experimental results show that they appear to emanate from the spiral layer locking, and the Mok signal is inverse-linearly dependent on the gate voltage. Figure 16c shows the RMCD signal of the vertical magnetic field double-layer CrI_3_ device (device 1) at zero gate voltage. Li et al. [290] demonstrated that the coupling of dark excitons and optically silencing chiral phonons makes the intrinsic photoluminescence of dark exciton copies in monolayer WSe_2_ possible, as shown in Figure 16d. The effect of the top gate voltage and magnetic field on the PL spectrum and excitation power of the device was investigated by encapsulating a single layer of WSe_2_ with a top gate electrode in contact with graphene (Figure 16e,f). Similarly, Xing et al. [291], reported the identification of Cr_2_Ge_2_Te_6_ ferromagnetic flakes (CGT) of thickness as low as a few nanometers, and the single-layer CGT diagram is shown in Figure 16g. In the magnetic field of 0.1 t along the ab surface of the crystal, when the temperature drops below~61 k, the magnetization intensifies (Figure 16h). This further demonstrates the enormous modulation of channel resistance of a 2D CGT device through the electric field effect. The experimental data further show that the 2D lattice voltage is adjustable and ferromagnetic 2D material CGT can be used as a new quantum functional material. Figure 16i shows an optical image of an 8.5 nm CGT wafer. Further cognition of magneto-optical mechanism and the design to control 2D material structure, 2D material-based devices will show more extensive applications. Thermo-optic modulation is another commonly used effective approach to realize high performance modulator devices, which rely on temperature-dependent refractive index changes of a certain material. Generally, the refractive index of materials is varied through the heating process. Then, the incident light passed through the employed optical waveguide filled with 2D materials, and the phase modulation via a resonant structure or interferometry can be achieved. As aforementioned, 2D materials possess superior properties, such as high electrical, thermal conductivity, and wide optical absorption region, which enable 2D materials to hold great potential for thermo-optic modulation applications. However, most reported investigations of thermo-optic approach are concentrated on visible to NIR band. To meet the requirements of different applications, it is of great significance to develop 2D materials-based MIR thermo-optic modulators, such as BP, Te, novel TMDCs, and so forth.

Acousto-optic modulators use sound waves to change a material’s refractive index, so that light diffraction and frequency variations are controlled. Acousto-optic modulators based on various 2D materials have been extensively studied in optical communication, display pulse generation (Q switch), and signal modulation. Lithium niobate has unique piezoelectric and birefringent properties, but its application in the field of optoelectronics is restricted by light activity and semiconductor transmission. Preciado et al. [292] manufactured and characterized the hybrid MoS_2_/LiNbO_3_ acousto-electric device using an extensible route and photolithographic definition of the FET structure using the millimeter MoS_2_ at the top (Figure 17a). The prototype device represents a device on silicon that competes with the electrical properties of MoS_2_. Surface acoustic waves excited on the substrate can be operated and detected in a contactless manner in a single layer of equipment. Figure 17b shows the FET emitted in the channel region by 400 m × 350 m spatial graph PL of characteristic monolayer MoS_2_. Recently, the saturable absorber is introduced with a new Tm:Ca (Gd, Lu) AlO_4_ dual Q-switched laser based on an acousto-optic modulator (AOM) and WS_2_ [293]. Under these conditions, the modulation rate (MR) of Tm:Ca (Gd, Lu) AlO_4_-based dual Q-switched laser has 1 kHz AOM, 91 ns pulse width, and 1.2 kW maximum peak power.

Compared with WS_2_ or AOM, pulse width compression ratio can reach to 15.38, corresponding to the maximum peak power of 511.30. The experimental data proves that WS_2_ has a satisfactory saturation absorption, and the double-Q switching laser is desirable to compress pulse width and increase peak power. Figure 17c–e show that the different Q-switched laser pumps of typical pulse train and corresponding mono pulse is 7.2 W, which confirms that the pulse amplitude of the double-Q switched laser is more stable compared with the single passive Q-switched laser. Chizhikovet et al. [294], developed an acousto-optic (AO) modulator made from bismuth sodium NaBi (MoO_4_)_2_ crystals. Isotropic orthogonal interaction geometry is used, longitudinal piezoelectric plate is used as piezoelectric transducer, and the sound wave is oriented along the crystal axis Z. The modulator shows high diffraction efficiency (up to 87%), 2.5 W RF power. It is characterized by manufacturability in terms of crystal growth, machining, and optical uniformity.

Huang et al. [295], proposed an electro-absorption modulator with indium phosphide. Insulating silicon wafers used as the substrate of the proposed electro-absorption modulator, as well as adhesives and indium phosphide installed on the insulating silicon wafers, as demonstrated in Figure 18a. The performance of the electro-absorption modulator was measured, as shown in Figure 18b. When the voltage is positive, the exciton transition causes a large change in the absorption change. The spectrum measured in the same condition is shown in Figure 18c. The figure shows that the driving voltage is relatively low when the bias voltage is positive. The electro-absorption modulator achieves 1.25 Gbps modulation with a driving voltage of 50 mV. This work confirmed that indium phosphide can be used as a low voltage driving modulator material. For improving the photo response liveness of lithium niobite materials, Preciado et al., combined MoS_2_ and lithium niobite (MoS_2_/LiNbO_3_) for acousto-electric device [292]. LiNbO_3_ was used as the substrate, and MoS_2_ was placed on the substrate. In addition, four metal electrodes are grown on the device by chemical deposition, as shown in Figure 18d. The insets on the left and right represent the measurement methods of four contacts and two contacts, respectively. When the voltage between the drain and the source is zero, the device resistance and conductance change with the voltage between the gate and the source, as is shown in Figure 18e. Figure 18f demonstrates the reaction activity of the two channels with time. Among them, red light and infrared laser are irradiated on the device at 6 s. It can be seen from the figure that the response speed of the two methods are very fast when lasers start to work. These findings confirmed the potential of MoS_2_/LiNbO_3_ as an infrared acousto-electric device.

## 5. Photodetectors

The development of new material synthesis technology, heterogeneous junction production method and micro-nanometer scale device processing technology, with excellent photoelectric properties has promoted the rapid development of 2D material photoelectric detection research [296]. Amani et al., used a cavity substrate with Au/Al_2_O_3_ layers as a back-reflector and λ/4 spacer, respectively, in combination with narrow gap Te (body 0.35 eV and 1 eV in 1 L) to enhance infrared light absorption, as shown in Figure 19a [156]. The cutoff wavelength is widened from 2.0 μm to 3.4 μm, as shown in Figure 19b,c. Multiple internal reflections of an incident laser in a plane cavity resulting in the fast rapid response of 16 A W^−1^ at V_d_ = 5 V for the device on 150 nm Al_2_O_3_ at λ = 1.7 μm. Huo et al. demonstrated a mixed sensitized mercury telluride/titanium dioxide/molybdenum disulfide PD (Figure 19d) [297]. Titanium dioxide (3.2 eV) was selected as the electron acceptor buffer layer to support charge transfer to the molybdenum disulfide channel, as shown in Figure 19e. Figure 19f indicates the spectral coverage of the hybrid structure with a detection wavelength of more than 2.0 μm. After optimizing the back-gate, the response of the device at 2 μm is about 500 A W^−1^, and the high D* is 10^12^ Jones (0.35 μW cm^2^, V_d_ = 1 V, V_g_ = −15 V). Unlike noble metal nanoparticles (typically gold or silver), semiconductor nanoparticles typically resonate in near-infrared or MIR [298,299,300]. Ni et al., demonstrated LSPR-based MIR hybrid PD coupled with graphene by borosilicate quantum dots, as shown in Figure 19g [301]. The LSPR effect of quantum dots introduced a strong local electric field to stimulate the absorption effect of graphene at a molecular resonance region. Figure 19h shows the simulated distribution of the square electric field |E|^2^ at the quantum dot/graphene, and it can be seen that LSPR is the strongest at 3μm. Through a series of studies, it has been found that combining 2D infrared materials with optical designs, such as quantum dots, can significantly improve performance and achieve high stability and high sensitivity of broadband photodetectors at required wavelengths.

### 5.1. Waveguide-Based Photodetectors

The BP band gap is narrow and limited, which is often used to improve dark current excess in graphene photodetectors. A gated multilayer BP detector integrated in a silicon photonic waveguide operating in the near-infrared telecommunication wavelength is demonstrated [302], as shown in Figure 20a. The BP detector has shown excellent performance and can work under small bias dark current. The intrinsic responsivity reached 135 mAW^−1^ and 657 mAW^−1^ in devices 11.5 nm and 100 nm thick, respectively (Figure 20b,c). Photovoltaic effect played a major role in photocurrent, whose response bandwidth exceeded 3 GHz. In order to further develop the integrated waveguide sensing system for MIR, Li et al. [303] realized the integration of silicon on insulator (SOI) waveguide and BP PD, as shown in Figure 20d. When working near the BP cutoff wavelength with weak absorption, the absorption length limitation of BP thickness limitation can be overcome by using the optical limitation of Si waveguide and grating structure, and the light–BP interaction can be enhanced. The response degree and noise equivalent power (NEP) of devices with different BP crystal orientations and thicknesses were discussed. In addition, the power-related responsivity and gate-adjustable photocurrent were also investigated. Under the bias of 1 V, the response of BP PD was 23 A/W at 3.68 μm and 2 A/W at 4 μm (Figure 20e,f). By integrating passive silicon photonics with active BP photodetectors, a new idea for MIR photodetectors is provided. In the same year, Deckoff-Jones et al. [304] developed a halcogenide glass waveguide-integrated BP MIR PD and manufactured devices along different crystal axes of BP for study. The influence of in-plane anisotropy on the optical response of waveguide integrated components is investigated. Devices A and B were made from 32.4 nm and 8.3 nm thick and thin sheets parallel to the armchairs, respectively. The response rate of the best equipment can be increased to 40 mA/W (Figure 20g,h), and the noise equivalent power at 2185 nm can be reduced to 30 pWHz^−1/2^ (Figure 20i). In addition, the responsivity of the PD changes by an order of magnitude in different BP directions. Therefore, BP has been proved to be a feasible medium infrared partial discharge material. Recently, Ma et al. [305], proposed a shared BP photonic system to achieve high responsiveness in a miniaturized BP waveguide photodetector. Near the edge of the band at about 3.8 μm, the responsiveness of the BP PD was tenfold higher on 10 μm long PhCWG than on subwavelength grating waveguides. Under the condition of the bias voltage of 0.5 V, BP PhCWG PD achieved 11.31 AW^−1^ response degree and 0.012 nW Hz^−1/2^ noise equivalent power.

2D materials other than BP can also be waveguide-integrated into photodetectors. Han et al. [306], reported an on-chip polycrystalline PbTe photoconductive detector with a sulphide glass waveguide through experiments (Figure 21a,b). At room temperature, the device showed a response of 1.0 A/W at wavelengths between 2.1 and 2.5 μm. In addition, the photoconductive signals of PbTe films in the wavelength range of 0.8–5 μm were measured. The response wavelength dependence of sample 1 under 10 V bias is displayed in Figure 21c. Liu et al. [307] used Ge-rich SiGe waveguides and transient components based on the light guide mode to conduct MIR sensing within the wavelength range of 5.2 and 6.6 μm (Figure 21d–g). For further verification, the absorption spectra of the independent photoresist spun onto a helical Ge-rich SiGe waveguide were monitored. At the spectral window of 5.8–6, the optical loss of the waveguide increases significantly. The 2 μm was identified and associated with intrinsic photoresist absorption. The platform’s ability to detect small amounts of methane gas is also discussed. Zhang et al. [308] theoretically investigated a tensile strain GeSn waveguide integrated with Si_3_N_4_ lining stress source for use in MIR detectors and modulators. Under tensile strain, the direct band gap of GeSn can be greatly reduced by reducing the conduction valley in the energy and increasing the degeneracy of the high band. GeSn waveguides absorption coefficients with different Sn compositions can be obtained accordingly. The cut-off wavelengths of the three different waveguide photodetectors can be extended to 2.32, 2.69 and 4.06 μm, respectively, as shown in Figure 21h–j.

Based on mature electronic equipment manufacturing technology, Younis et al. [309] combined SOI and Ge for MIR detectors. The 200 nm thick SOI is used as the substrate, and methods, such as epitaxial growth and chemical deposition, are used to obtain SiO_2_ flakes. In addition, 200 μm nano-cones placed on two sections of the waveguide, and the Ge-on-SOI proposed by the researchers are demonstrated in Figure 22a. Figure 22b demonstrates that the normalized transmission data of the rib waveguide. When the Ge thickness is 0.85 μm, the refractive index of the waveguide is shown in Figure 22c. The Ge-on-SOI proposed in this work has low transmission loss, and it can be used in MIR sensor applications. Zhang et al. [308], combined GeSn with silicon nitride for MIR detectors. Among them, Si is used as the substrate material, and silicon oxide, silicon, GeSn, and silicon nitride constitute the basic structure of the SOUP waveguide, as shown in Figure 22d. This structure can cause change in the waveguide through the stretching of silicon nitride. In the experiment, the relaxed and stretched states of the waveguide are compared. The band gap of the waveguide in the relaxed state is significantly larger than that of the stretched waveguide, as shown in Figure 22e. The change in the absorption spectrum is due to the change in the Sn content in GeSn as shown in Figure 22f. The cut-off wavelength of Ge0.90Sn0.10 is in the MIR range. This confirms that stretched GeSn can be used for waveguide MIR detectors.

### 5.2. Schottky Photodetector and Phototransistor

The rich physical characteristics of BP and its potential applications in the construction of nano-electron and nano-photonic devices have attracted great attention [310]. Yuan et al. [100] found that the working wavelength range of photonic devices could be greatly expanded by introducing arsenic into the black arseno–phosphorous alloy formed by BP (B-As_x_P_1−x_), as shown in Figure 23a,b. The prefabricated b-As_0.83_P_0.17_ photodetectors embedded in hexagonal boron nitrates (hBN) showed peak non-intrinsic responsibility—190, 16 and 1.2 mA/W at 3.4, 5.0 and 7.7 μm, respectively. Since the original b-As_0.83_P_0.17_ is preserved through complete hBN packaging, the inherent photoconductivity effect led the photocurrent generation mechanism, and the transmission hysteresis of these b-As_0.83_P_0.17_ photodetectors can be ignored. Therefore, the b-As_0.83_P_0.17_ alloy is an advancing material due to its wide optical responsiveness in the MIR range due to inherent photoconductance. Tan et al. [311], synthesized a new 2DM black carbonized phosphorus (b-PC) in order to achieve the responsiveness and response time required for both weak signal and high-speed detection (Figure 23c,d). The absorption spectrum can be as high as 8000 nm. Under the excitation wavelength of 2 um, the adjustable response and response time of b-PC phototransistor are shown in Figure 23e. The peak response rate of b-PC phototransistor is 2163 AW^−1^ and the equivalent power of granular noise at 2004 nm is 1.3 fW Hz^−1/2^. In addition, it showed that the response time of 0.7 ns can be adjusted by gating effect, which makes it universal for high-speed applications. A high performance antenna with integrated BP photoconductor with ultra-wideband detection from infrared to terahertz frequencies is demonstrated by Wang [312] ((Figure 23f–g). Different photoconductive mechanisms, such as photo thermoelectricity (PTE), radiant heat, and electron hole generation, can be designed due to the device geometry, input central wavelength, and power. In particular, when the photonic energy increased to the THz band, there is still a photoconductive response at room temperature.

Zhou et al. [313], demonstrated a new model of highly sensitive Schottky barrier-controlled phototransistor. It adopts a low layer BP channel perovskite modified by MAPbI_3−X_Cl_X_ (Figure 24a,b), and the channel current is affected by the Schottky barrier at the source electrode. In the perovskite layer near Schottky barrier, the electric field assisted the electron capture process, so the optical response velocity of the unit could be adjusted by varying the drain voltage. The device is capable of displaying a high responsiveness of 10^6^–10^8^ AW^−1^ (Figure 24c), an ultra-high ratio detection rate of up to 9 × 10^13^ Jones, and 10 ms response time.

For the purpose of developing technology in the field of Schottky phototransistor, Wu et al. [314], applied MoS_2_ to phototransistors. The Au–Cr–Au layered structure is placed on the top of MoS_2_, and this design can be used to distinguish the response mechanism. Under the illumination of 454 nm to 1550 nm, MoS_2_ transistor will show different light response properties. As shown in Figure 25a, the light response of different wavelength bands is displayed. In the infrared light band, the photocurrent signal is negative, and the response time is short. Figure 25b shows the change in scale transfer with light power. In addition, it is found in the experiment that the negative photoelectric signal value in the infrared band is related to the influence of ambient temperature. The transistor under low temperature and dark conditions is measured, as shown in Figure 25c. Among the figure, the inset shows the change of carrier mobility with temperature. This work can be applied to infrared light detection technology. The number of layers of TMDCs can affect the nature of the energy band gap. Ko et al. [315], used MoSe_2_ for phototransistor control. The Figure 25d demonstrates the Raman spectrum curves of MoSe_2_ with different layers are displayed. In the experiment, MoSe_2_ was placed on the Ti electrode by a mechanical peeling method, and the material of another electrode was Cu. The external electron efficiency of the MoSe_2_-based detector is demonstrated in Figure 25e. The figure shows that the maximum external electron efficiency of the MoSe_2_ PD can reach 37,745%. In addition, the researchers found that the photo response rate is opposite to the incident laser energy, as shown in Figure 25f. This is caused by the trapping effect of impurities, such as oxygen, on the surface of the material. The phototransistor proposed in this work can be applied to detectors, sensors and other devices in the near-infrared field. To improve the performance of phototransistor, Shi et al. [316], combined indium phosphide–zinc sulfide and MoS_2_ to obtain a hybrid Schottky transistor (InP@ZnS-MoS_2_). The electrode material is Pt, which can form a Schottky junction with MoS_2_, as shown in Figure 25g. This design can effectively improve the light response intensity and speed of the transistor. Among them, as the power density change, the light response can reach to 1374 A∙W^−1^, as shown in Figure 25h. In addition, the light response period of this design is shown in Figure 25i, and the self-powered speed of this transistor can reach 21.5 μs. The InP@ZnS-MoS_2_ transistor proposed in this work can improve the performance of the phototransistors effectively.

### 5.3. Photoelectrochemical Photodetector

The 2D BP nanocrystals were prepared by a convenient liquid stripping method. Ren et al. [317] prepared BP nanocrystals for the manufacture of self-powered photodetectors with good light response activity and environmental stability, as shown in Figure 26a,b. Photoelectric chemistry (PEC) test showed that under light irradiation, the current density of BP nanosheet could reach 265 nA cm^−2^, while the dark current density fluctuated around 1 nA cm^−2^. Furthermore, the device maintained excellent on/off performance even after one month of operation, as shown in Figure 26c. In addition, the PEC performance of BP nanoplate-based photodetectors was studied at various KOH concentrations, indicating that the BP nanoplate-prepared photodetectors may have great application potential in self-powered photodetectors. However, the inherent instability of BP limits its application in optoelectronic devices. Therefore, Zhang et al. [318], used the hydrophobic polyionic liquid poly (1-hexyl-3-vinylimidazolium) hexafluorophosphate (PIL-TFSI) to encapsulate BP quantum dots to form BP-PIL (Figure 26d) and applied it to a photoelectrochemical photodetector (PD). From the experimental results and density functional theory, it is found that the stability of BP significantly improved and the fluoride of BP is significantly enhanced. The prepared PD showed significantly improved optical response behavior (542 nA cm^−2^) and remained unchanged after 90 days, as shown in Figure 26e,f. In addition, the properties of PIL-TFSI enable the self-healing ability of the prepared PD, and the ON/OFF signal still performs well after 50 cycles. Qiao et al. [319], obtained the hetero junction of BP QDS-molybdenum disulfide (BP QDS-MOS_2_) and set up a PEC PD-based heterostructure (Figure 26g). The PEC PD with BP QDS-MOS_2_ heterojunction has significantly enhanced the optical response performance, as shown in Figure 26h,i. At the same time, the PEC type PD also shows the light response behavior from the power supply, good light response performance, and stability in the liquid environment. The results prove once again that using BP quantum dots as hole receptors is a feasible method to promote the separation of photoelectric hole pairs.

Indium selenide (InSe), as a typical III-VI layered material successfully catches extensive attention by way of its unique layered structure and wide adjustable band gap. Li et al. [320] proposed a small layer InSe nanosheet, which has a direct band gap obtained by a mild liquid phase stripping method and was used to construct a photochemical (PEC) type PD (Figure 27a,b). The detector shows good current density, light response, and cycling stability in the KOH solution (Figure 27c). The detection performance of PEC InSe PD can be changed with the change of solution concentration and applied voltage, indicating that it can be used as a potential candidate for PD. In addition, the extended optimization of the photoelectric chemical properties of InSe nanosheets will be further applied to other equipment, such as sensitized solar cells, water decomposition systems, and optical tracking systems. In order to find a high quality photoelectrode for water decomposition and water absorption, Ren et al. [321] synthesized a new type of plasma photoanode heterostructure which consists of plasma Ag and fullerene shell-WO_3−X_ (Figure 27d–g). Among them, the hot electrons generated by the exciter of Ag nanomaterials could be effectively transported to WO_3−X_ nanosheet. The conversion efficiency and photodegradation of PEC were improved due to the presence of silver and fullerene shell-WO_3−X_. Meanwhile, photovoltaic devices photoelectric conversion efficiency can be further improved by increasing concentrating modules [322,323,324,325,326].

ReSe_2_ is a 2D TMDC material, which has a hexahedral structure, and the structure is different from the hexahedral structure of other TMDCs materials. The ReSe_2_ has different properties in various directions. In order to apply this recently discovered material to PDs, Yang et al. doped Mo into ReSe_2_ material to obtain an octahedral semiconductor material (Mo: ReSe_2_) [327]. The Raman spectrum of Mo: ReSe_2_ is demonstrated in Figure 28a. The relationship curve in the figure shows the Raman forms of Mo: ReSe_2_ is more than to other TMDCs. In the experiment, silicon oxide is used as the substrate and Mo: ReSe_2_ material is placed on the silicon oxide. The Au material is selected as the electrode, as shown in Figure 28b. Meanwhile, the researchers found that the performance of the Mo: ReSe_2_ material in NH_3_ after annealing is better than that in normal environment, as shown in Figure 28c. Under 633 light, the external electron efficiency of Mo: ReSe_2_ in ammonia gas can reach 10893%. This work proves that ReSe_2_ can be used as an excellent photoelectric sensor material. MoS_2_ materials are easy to prepare and have a low cost, but its electron migration rate is lower than that of silicon crystals. For the purpose of improving the shortcoming of MoS_2_, Li et al. obtained a material (SrTiO_3_@ MoS_2_) for PDs by combining SrTiO_3_nano material and MoS_2_, as shown in Figure 28d [123]. In the experiment, SrTiO_3_ was used as the substrate and MoS_2_ loaded on the substrate. The X-ray diffraction images of several materials are shown in Figure 28e. Among them, the green, purple, and orange lines represent MoS_2_, SrTiO_3_ and SrTiO_3_@ MoS_2_ materials, respectively. The figure shows that the X-ray diffraction peak of MoS_2_ is very low, while the peak of SrTiO_3_ is more obvious. The current density of SrTiO_3_ and SrTiO_3_@ MoS_2_ materials changed with time in the KOH electrolyte are shown in Figure 28f. Among them, the peak current density of SrTiO_3_@ MoS_2_ is 21.4 μA, which is much higher than that of SrTiO_3_. This work has contributed to the development of MoS_2_ composite materials. To make the mass applications of TMDCs materials served as PDs possible, Patel et al. [328] combined WSe_2_ and MoSe_2_ to form a heterojunction structure (MoSe_2_-WSe_2_). In the experiment, the Si substrate is used as the base, and the MoSe_2_ film is placed on the base. After that, WSe_2_ is placed on the top layer, as demonstrated in Figure 28g. The Raman spectrum of MoSe_2_-WSe_2_ material is shown in Figure 28h. In addition, the light response of the material will change with the change of light intensity, as shown in Figure 28i. Compared with the traditional TMDCs/Si structure, the MoSe_2_-WSe_2_ structure has better light response properties. This confirmed the application potential of the large-area TMDCs in the optical field.

TIs can be used as materials for photoelectrochemical PEC, Zhang et al. combined Bi_2_Se_3_ and Te@selenium (Te@Se) for PEC device [329]. In the experiment, Te@Se nanomaterial was used as the substrate, and Bi_2_Se_3_ was grown on the substrate by hydrothermal method. Finally, the Bi_2_Se_3_/Te@Se heterostructure required for the experiment was obtained, and the reaction process is shown in Figure 29a. The X-ray diffraction spectra of Se, Bi_2_Se_3_ and Bi_2_Se_3_/Te@Se were measured, as shown in Figure 29b. Among them, due to the problem of growth uniformity, the peak value of the Bi_2_Se_3_ material obtained by epitaxial growth has shifted. When the wavelength is constant, the photocurrent density of Bi_2_Se_3_/Te@Se in different alkaline solutions is different, as shown in Figure 29c. The figure shows that in the HCL solution, Bi_2_Se_3_/Te@Se material has better self-driving ability. In addition, the stability of Bi_2_Se_3_/Te@Se in several alkaline solutions has been tested. The Bi_2_Se_3_/Te@Se material still has relatively stable reaction ability after one month. This work proves that Tis can be used in PEC devices. In order to optimize the performance of Tis in PEC light detector, Ren et al. compared the Au-Bi_2_Te_3_ substance solutions of different concentrations [330]. The Au-Bi_2_Te_3_ concentrations of 1%, 3% and 5% are measured in alkaline solution, respectively. The SEM image of 3% Au-Bi_2_Te_3_ is shown in Figure 29d. X-ray diffraction is used to characterize several materials, as shown in Figure 29e. The X-ray diffraction curves of pure Bi_2_Te_3_ materials, 1%, 3%, and 5% Au-Bi_2_Te_3_ are compared. It can be seen that the peaks of several metal-semiconductor materials are consistent with the peaks of pure Bi_2_Te_3_ materials. Among several materials, the light response intensity and photocurrent of 3% Au-Bi_2_Te_3_ are better than other materials. Figure 29f shows the photocurrent intensity curve of the material with different bias voltages. The inset is the responsivity and light conversion efficiency. This work improves the light response performance of TIs materials by combining Au particles and Bi_2_Te_3_, and researchers confirmed the potential of Au-Bi_2_Te_3_ materials in a PEC detector.

To facilitate a better understanding of the 2D materials based photodetector applications, a comparison of photodetection spectral range and other key parameters of photodetector is listed in Table 4.

### 5.4. MIR Imaging

The imaging systems usually require minimization, multifunction, adaptability, and good controllability in different environments. Lei et al. [160], uses 2D Te materials to study stable MIR polarization imaging as shown in Figure 30. Wide band sensitive light response in ambient temperature shows excellent stability, and it will not degrade in the atmospheric conditions. These findings show that the anisotropy of 2D Ti ensured the polarization imaging in the scattering environment, and the linear polarization degree exceeds 0.8, which expands the idea of polarization medium infrared imaging technology.

## 6. Results Conclusions and Outlook

Since the extraordinary optoelectronic properties, 2D materials beyond graphene are intensively employed to fabricate MIR optoelectronic devices. In this review, we investigated the recent progress of 2D materials-based MIR optoelectronic devices, including 2D material candidates that are suitable for MIR applications, such as BP, TMDCs, GDY, 2D Te nanoflakes, perovskites, and Topological insulators. As well as their device application in MIR bands, in particular, light emission devices, modulators, and photodetectors.

In terms of LEDs, compared to commercially available LEDs, the external quantum efficiency and operation stability of 2D materials-based LEDs are still too low. To satisfy the demand of practical applications, the external quantum efficiency and operation stability need to be significantly improved. Furthermore, the emitting wavelength needs to be further extended, which is far from the practical application. For driven mode, more attention should be concentrated on the electrically driven mode. Thus, it is of great significance to develop novel 2D materials-based LEDs with various configurations in the future.Regarding single-photon emitters based on 2D materials, they are thought to be originated from the defects. However, the underlying physical mechanism, excitation processes, and atomic structure are still under debate. Meanwhile, the emitting wavelength needs to be extended into the deep MIR region.The lasing threshold is relatively low at a lower temperature, however, for room temperature lasing, the threshold needs to be significantly suppressed.For modulator applications, the irradiation damage threshold of 2D materials needs to be improved. In addition, the long term operation stability needs to be enhanced as well, which plays a determining role for practical applications. Moreover, only a few kinds of 2D material can be utilized for electro-optic modulators. Further improving the architecture of optical modulators and exploring novel 2D materials may provide an alternative means to overcome these challenges.In terms of MIR photodetectors, since the weaker optical absorption of the MIR light, the photo responsivity, carrier mobility, and response speed are much lower and slower than that of visible and NIR photodetectors. Combined with other materials with higher MIR light absorption coefficient to establish heterojunction may provide an effective way to solve these problems.For MIR imaging applications, the recent devices mainly adopt a point photodetector, which is far from the practical application. MIR imaging devices based on large scale 2D material arrays should be developed.

In conclusion, 2D materials beyond graphene-based MIR optoelectronic devices have already achieved some milestone achievements. However, these devices also face some severe challenges as previously mentioned. With continuous investigation, we believe that a more comprehensive understanding of MIR optoelectronic devices based on 2D materials beyond graphene will emerge in the near future as a result of these ongoing concerted research efforts.

## Figures and Tables

**Figure 1 nanomaterials-12-02260-f001:**
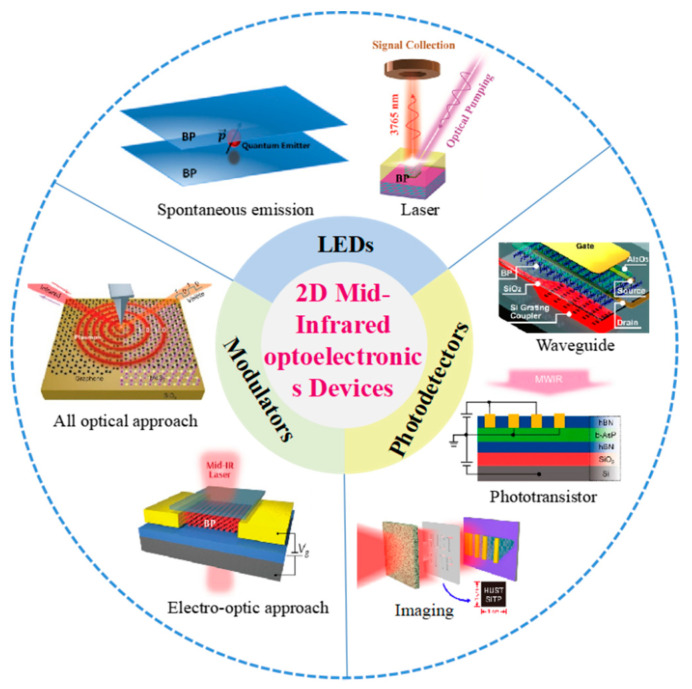
The 2D materials for various MIR optoelectronic applications.

**Figure 2 nanomaterials-12-02260-f002:**
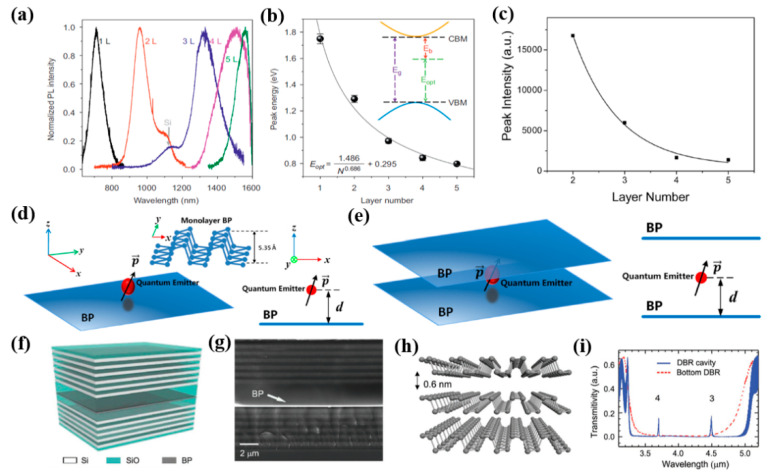
(**a**) Normalized PL spectra of BP from 1 to 5 layers. (**b**) PL peak energy of different layers [185]. Copyright 2015 Nature Publishing Group. (**c**) Relationship between peak intensity and number of layers [186]. Copyright 2014 American Chemical Society. (**d**) Single layer BP diagram coupled with quantum emitter. (**e**) Double-layer BP diagram coupled with quantum emitter [187]. Copyright 2017 The Optical Society. (**f**) Embedded DBR microcavity embedded with BP nanosheets. (**g**) The cross-section view of microcavity embedded with BP inSEM image. (**h**) BP crystalline structure. (**i**) DBR and the SiO cavity transmission spectrum caught in the bottom and up DBRs. (**f**–**i**) Reproduced with permission from Ref. [188]. Copyright 2020 Wiley-Blackwell.

**Figure 3 nanomaterials-12-02260-f003:**
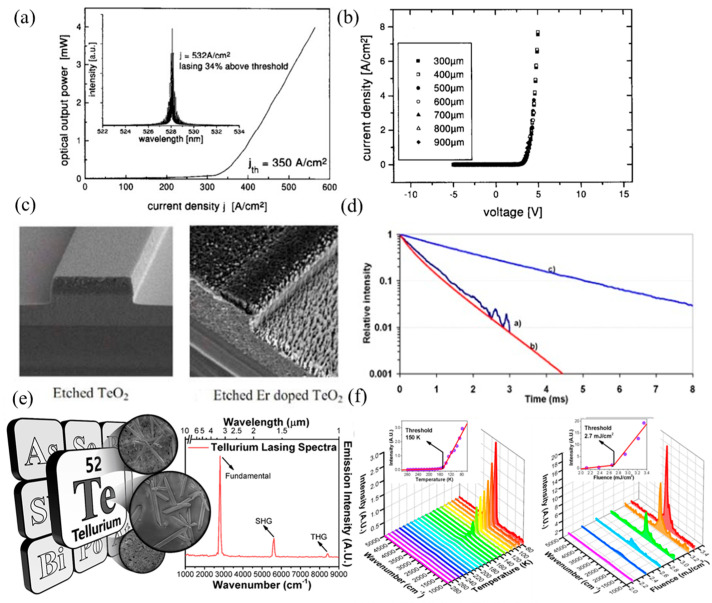
(**a**) Power/current curve and emission spectra of ZnSe laser diodes with Te-terminated GaAs interface (inset). (**b**) As the electric field changes, the Te-started laser diode is displayed [189]. Copyright 1998 Elsevier. (**c**) Etching TeO_2_ waveguide and Er-doped waveguide. (**d**) Life time variation of erbium-doped acetate film [190]. Copyright 2010 The Optical Society. (**e**) The photoluminescence of Te solid crystals in the medium wavelength infrared (MWIR) region. (**f**) Temperature variation curve of MWIR emission spectrum of Te bulk crystal, and the curve of Te bulk crystal MWIR emission spectrum with laser intensity [197]. Copyright 2019 American Chemical Society.

**Figure 4 nanomaterials-12-02260-f004:**
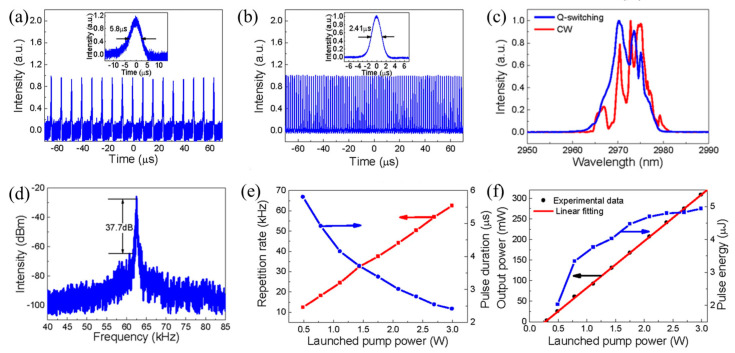
(**a**) Pulse series and monopulse waveform (inset) are 489.3 mW and (**b**) 2.99 W, respectively, at the power of the transmitting pump. (**c**) The measured spectrum of the pulse. (**d**) The RF spectrum of the pulse. (**e**) Change curve of launched pump power with repetition rate. (**f**) Launched pump power change curve with output power [205]. Copyright 2016 Nature Publishing Group.

**Figure 5 nanomaterials-12-02260-f005:**
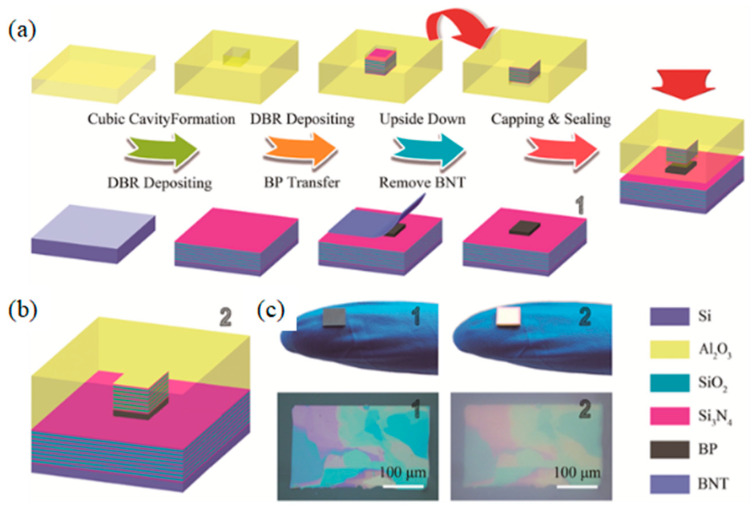
(**a**) Manufacturing process of BP laser is designed with an open cavity surface emitting laser device. Using sapphire as the substrate, DBR is deposited in the cubic step hole, bonding the substrate reverse. After preparing the steps for DBR deposition on the silicon substrate and bottom assembly, transfer the Nitto tape with blue Nitto tape, and finally combine them to obtain the laser device. (**b**) Schematic diagram of the completed laser device. (**c**) The image on the left shows the BP film covering the waveguide and the optical micrograph, while the image on the right shows the final image and the optical micrograph of the device [211]. Copyright 2019 American Chemical Society.

**Figure 6 nanomaterials-12-02260-f006:**
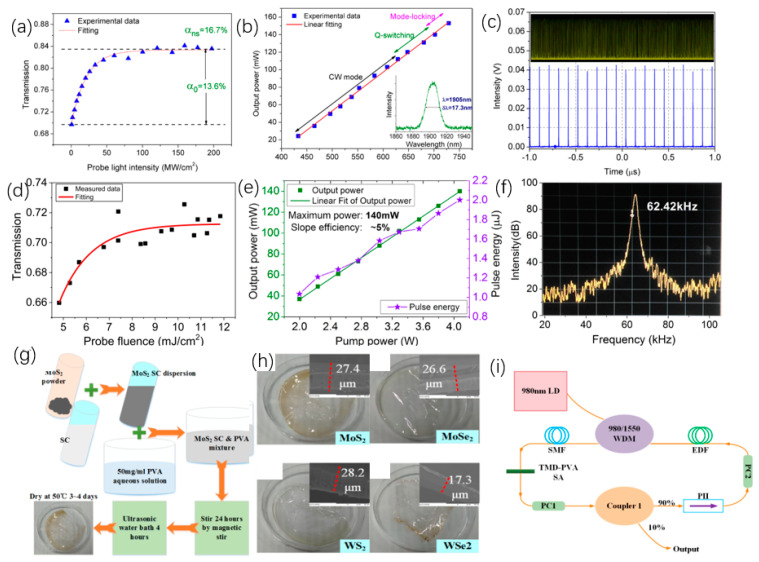
(**a**) Nonlinear absorption curve of MoS_2_ on the mirror. (**b**) The change curve of pump power and output power. (**c**) Pulse train diagram of the fiber laser [182]. Copyright2015 IOP Publishing LTD. (**d**) Measurement data of MoS_2_ as a saturable absorber. (**e**) Trend chart of output power, pulse power, and pump power. (**f**) Change of radio frequency spectrum with different pump power [213]. Copyright 2019 Institute of Physics Publishing. (**g**) Preparation method of MoS_2_-PVA film. (**h**) Four images of TMDCs–PVA. (**i**) Schematic diagram of the experimental structure with TMDCs–PVA as saturable absorber [214]. Copyright 2015 Optical Society of America.

**Figure 7 nanomaterials-12-02260-f007:**
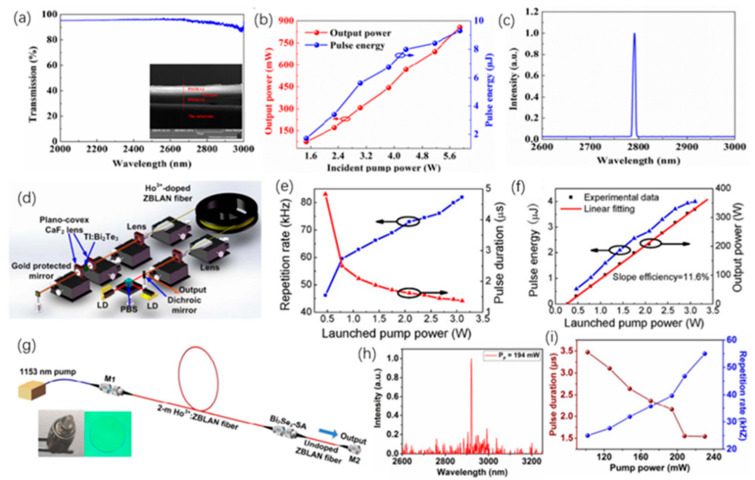
(**a**) Transmission spectrum of Bi_2_Te_3_ saturable absorber in the range of 2000–3000 nm. (**b**) The relation curve of output power, pulse energy change curve, and pump power. (**c**) Laser spectrum when the pump power is at the peak value [215]. Copyright 2016 Institute of Electrical and Electronics Engineers Inc. (**d**) Structure diagram of experimental platform of fiber laser. (**e**) When Bi_2_Te_3_ is used as a saturable absorber, the relationship between repetition frequency and pulse duration and pump power. (**f**) The relationship of pulse power, output power, and pump power [210]. Copyright 2015 Optical Society of America. (**g**) Structure diagram of experimental platform of Bi_2_Se_3_ fiber laser. (**h**) Output spectrum curve of the fiber laser. (**i**) The variation curve of pulse duration, repetition rate, and pump power [216]. Copyright 2018 IEEE.

**Figure 8 nanomaterials-12-02260-f008:**
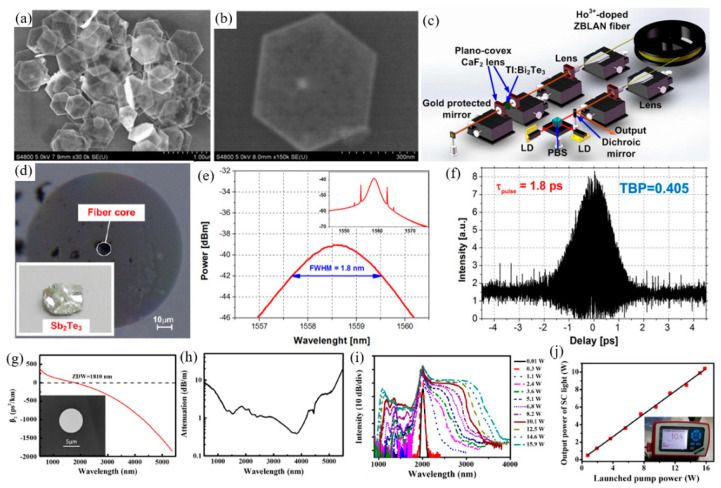
(**a**) Lower and (**b**) higher magnification SEM images of Bi_2_Te_3_ samples. (**c**) The device of passive Q-switched Ho^3+^-doped fiber laser based on the TI: Bi_2_Te_3_SA [210]. Copyright 2015 The Optical Society. (**d**) Morphology of Sb_2_Te_3_ layers under atomic force microscope. The central wavelength of the emitted solitary light is 1558.6 nm. (**e**) Spectrum with indication of 3 dB bandwidth. Illustration: spectral record of the 30 nm span, (**f**) 1.8 ps pulse Autocorrelation [225]. Copyright 2014 American Institute of Physics. (**g**) Dispersion curve of the fluortellurate glass fiber, inset: photo of the end face of the fluortellurate glass fiber. (**h**) Loss curve of fluortellurate glass fiber. (**i**) Evolution trend of supercontinuum laser output spectrum with pumping laser power. (**j**) Curve of output power of supercontinuum laser with pump laser power [236]. Copyright 2018 OSA Publishing.

**Figure 9 nanomaterials-12-02260-f009:**
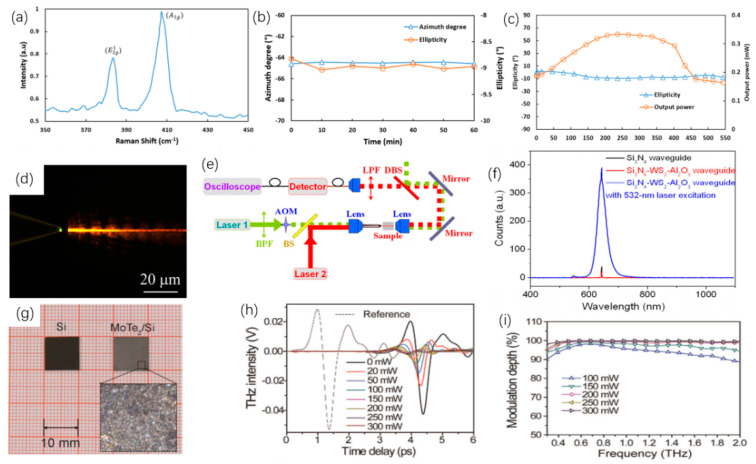
(**a**) Raman spectroscopic characterization of the prepared MoS_2_ film. (**b**) The change curve of the azimuth and ellipticity in a period. (**c**) The relationship curve of ovality, output power, and pump power [259]. Copyright 2019 Elsevier B. V. (**d**) The microscopic image of Si_3_N_4_-WS_2_-Al_2_O_3_ illuminated by a light source. (**e**) Structure diagram of the test platform. (**f**) Spectra of several modulator materials with laser excitation [261]. Copyright 2017 American Chemical Society. (**g**) Image of silicon substrate and MoTe_2_/Si. (**h**) THz spectrum with different power laser irradiation. (**i**). Modulation depth map with different laser power in 0.3–2.0 THz [261]. Copyright 2020 John Wiley and Sons Inc.

**Figure 10 nanomaterials-12-02260-f010:**
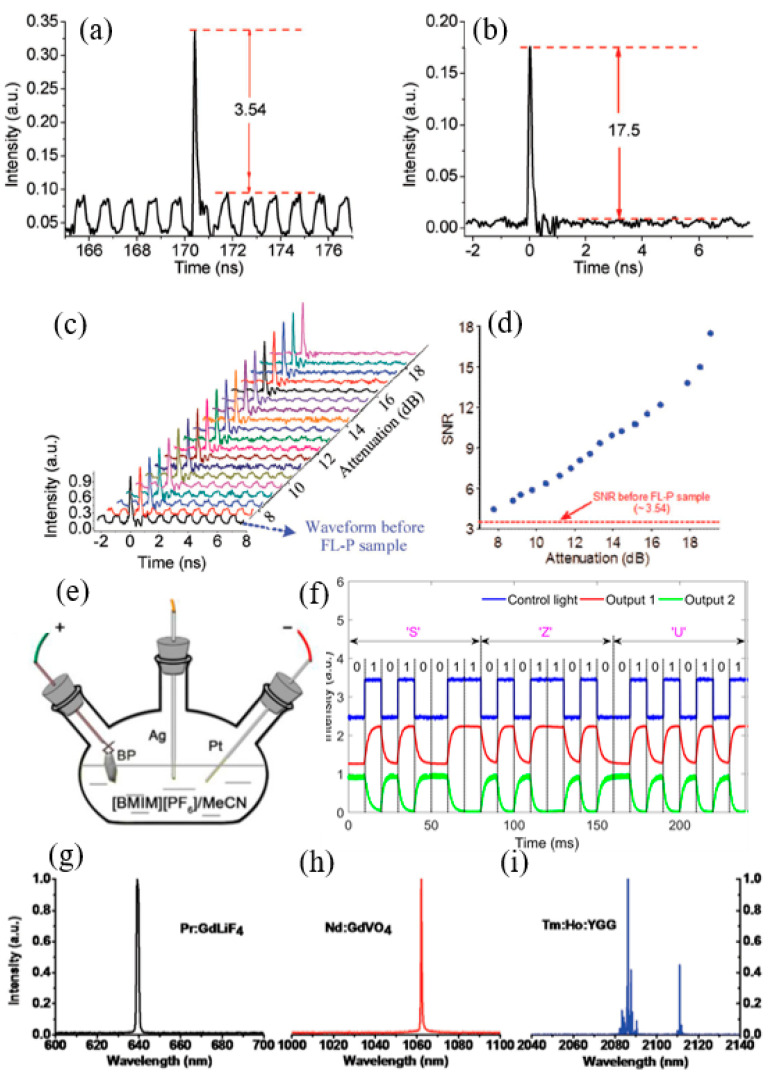
(**a**) Experimental results of all-optical threshold: the optical pulse waveform before entering the threshold device. (**b**) The optical pulse waveform after passing through the threshold device. (**c**) Time pulse waveform evolution diagram of different incident light power. (**d**) The corresponding signal-to-noise ratio at different incident optical power [262]. Copyright 2017 John Wiley and Sons Inc. (**e**) Schematic diagram of FP synthesis in an electrochemical stripping unit. (**f**) ASCII codes of ‘S’, ‘Z’, and ‘U’ obtained by control light and output ports [263]. Copyright 2018 Wiley-VCH Verlag. (**g**) 639 nm, (**h**) 1.06 μm, and (**i**) 2.1 μm passively modulated volume lasers, respectively [264]. Copyright 2015 John Wiley and Sons Inc.

**Figure 11 nanomaterials-12-02260-f011:**
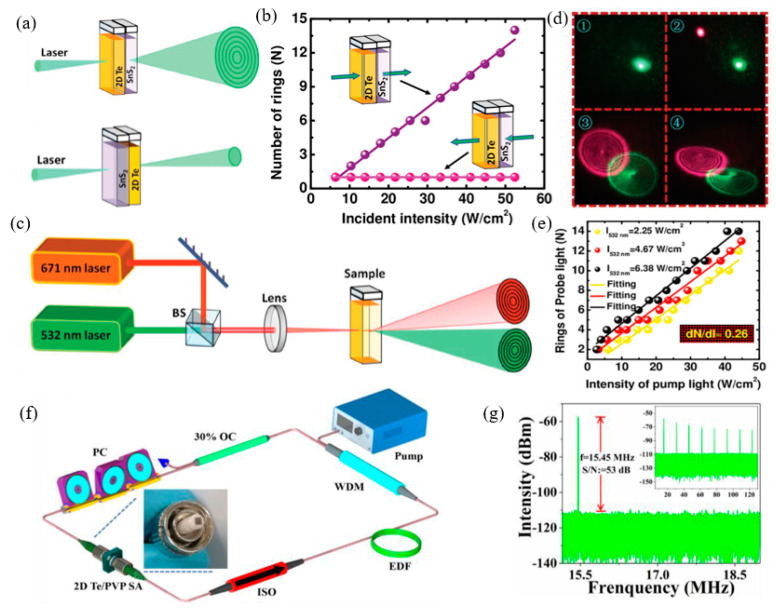
(**a**) The forward and reverse nonlinear responses of 2D Te/SnS_2_ nanocrystalline photonic diodes. (**b**) Results of non-reciprocal propagation of light. (**c**) Schematic diagram of all-optical modulation structure. (**d**) Modulation of pump light (red light) to detect light (green light). (**e**) With the change of the light intensity of the pump, the change of the number of optical diffraction rings is detected [266]. Copyright 2019 Wiley-VCH Verlag. (**f**) Schematic of the mode-locking fiber laser. (**g**) Radio spectrum [265]. Copyright 2019 The Royal Society of Chemistry.

**Figure 12 nanomaterials-12-02260-f012:**
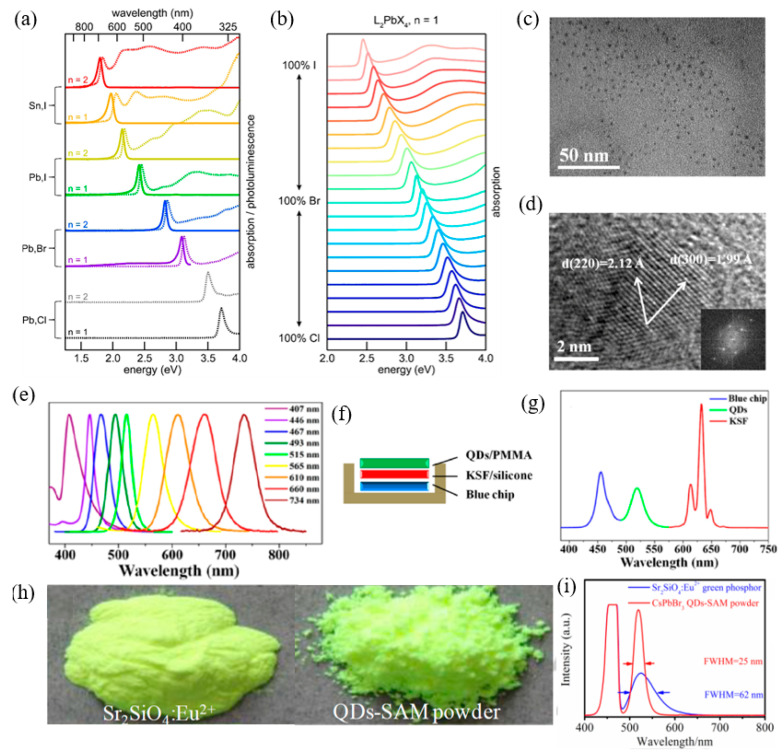
(**a**) When n = 1, n = 2 nanoplatelets were absorbed by solution phase and photoluminescence spectra. (**b**) The emission spectrum of 2D perovskite nanomaterials when n = 1 [273]. (**c**) TEM image of colloidal CH_3_NH_3_PbBr_3_ QDs. (**d**) The size distribution diagram of samples in figure (**a**). (**e**) PL emission spectra of CH_3_NH_3_PbBr_3_ QDs. (**f**) Schematic diagram and (**g**) EL spectra of pc-WLED devices based on green emissive CH_3_NH_3_PbBr_3_ QDs and red emissive rare-earth phosphor KSF [274]. Copyright 2015 American Chemical Society. (**h**) Optical images and (**i**) PL spectra of Sr_2_SiO_4_:Eu^2+^ green phosphor and CsPbBr_3_ QDs-SAM powder [275].

**Figure 13 nanomaterials-12-02260-f013:**
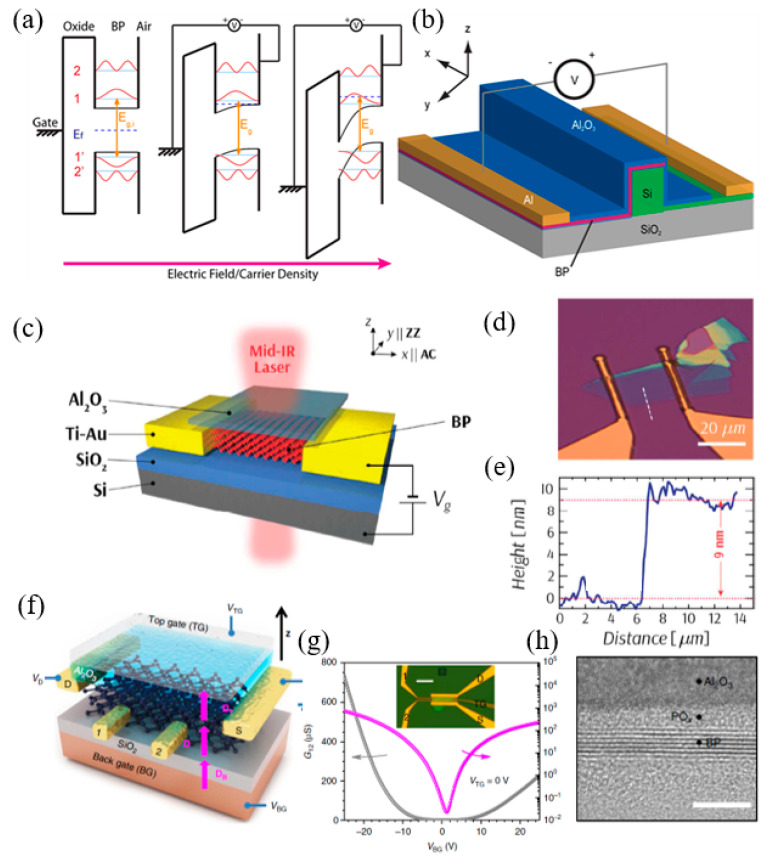
(**a**) Schematic diagram, band diagram, and wave function of 5 nm thick BP QW. (**b**) Modulator schematic [276]. (**c**) Schematic diagram of multilayer BP modulator. (**d**) Optical microscope image of BP modulator. (**e**) Atomic force microscope image of BP modulator [277]. Copyright 2017 American Chemical Society. (**f**) Test program for BP band gap tuning. (**g**) 4 nm thick BP film conductance as a function of top gate zero bias voltage and back gate bias voltage. (**h**) BP under an atomic microscope [278]. Copyright 2017 Nature Publishing Group.

**Figure 14 nanomaterials-12-02260-f014:**
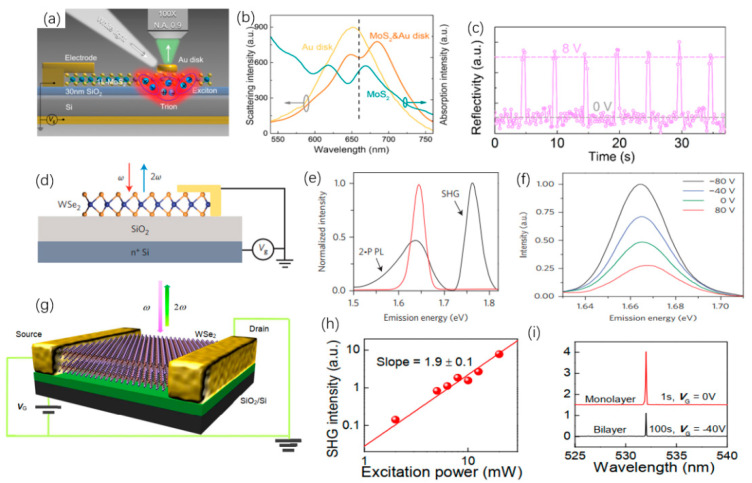
(**a**) Structure schematic diagram of Au/MoS_2_. (**b**) The scattering spectrum curve of gold dish and Au/ MoS_2_, and the absorption spectrum curve of MoS_2_. (**c**) Modulator switching time under an applied voltage with a period of 1 ms [279]. (**d**) Schematic diagram of WSe_2_ used as an electro-optic modulator. (**e**) The intensity curves of SGH and two-photo induced photoluminescence at different emission energy. (**f**) SGH spectrum curve under different gate voltage [280]. (**g**) Schematic diagram of double-layer WSe_2_ used in field effect transistors. (**h**) The relationship between incident light power and CHISHG intensity. (**i**) In the same experimental environment, the SHG intensity comparison chart of signal-layer WSe_2_ and double-layer WSe_2_ [281].

**Figure 15 nanomaterials-12-02260-f015:**
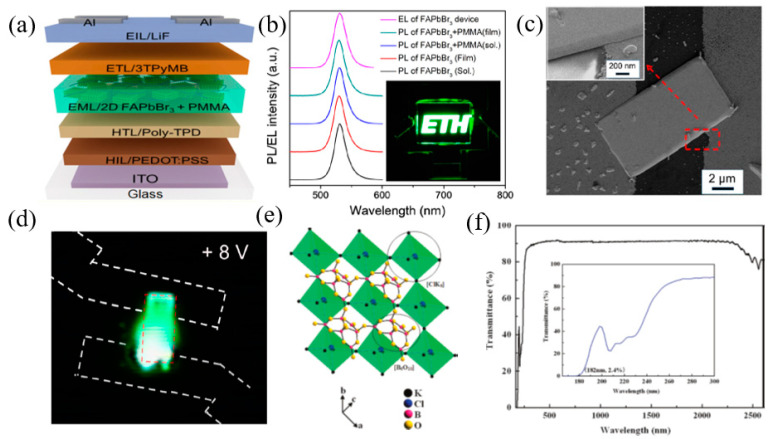
(**a**) Schematic architecture of vertical electroluminescent device. (**b**) PL spectrum of colloidal 2D FAPbBr_3_perovskite [284]. Copyright 2017 American Chemical Society. (**c**) SEM images of EL device showed that CsPbBr_3_ nanometer plates grown in gas phase were bribed on two ITO electrodes. Azo tilt image of nanometer plate (upper left image). (**d**) Under the condition of positive bias voltage V = +8 V, EL spectrogram of CsPbBr_3_ nanometer plate EL device of the upper electrode was obtained [285]. Copyright 2017 World Scientific Publishing Co. Pte Ltd. (**e**) The 3D diagram of KBOC with KO bonds. (**f**) KBOC crystal spectrum achieved from UV–vis–IR transmittance. The inset exhibits the transmittance versus λ curve between 165 and 300 nm [286]. Copyright 2011 American Chemical Society.

**Figure 16 nanomaterials-12-02260-f016:**
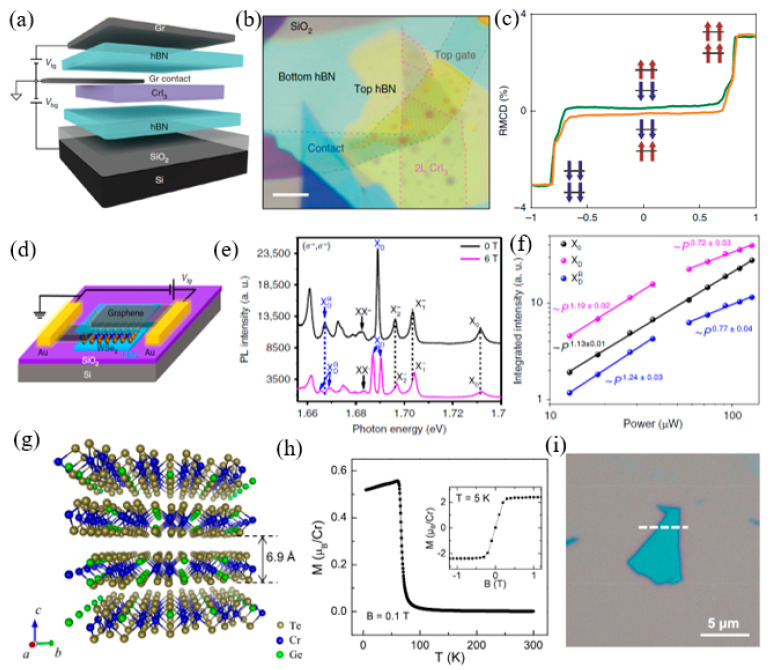
(**a**) Schematic diagram of a double-gated and double-decker CrI_3_ device assembled. (**b**) False-color optical micrograph. (**c**) Variation curve of RMCD signal with zero grid voltage [289]. Copyright 2018 Nature Publishing Group. (**d**) BN encapsulated single-layer WSe_2_ with graphene contact and top gate electrode. (**e**) The PL spectrum of the device at (**c**) 4.2 K. (**f**) Curve of integrated intensity and excitation power [290]. Copyright 2019 Nature Publishing Group. (**g**) The stratified crystal structure diagram of CGT along the direction of (0 0 0 1). (**h**) The magnetization of CGT single crystal as a function of the temperature change at 0.1 T magnetic field. Relationship between measured magnetization and magnetic field at 5 K (inset). (**i**) Classic 2D CGT chip optical image [291]. Copyright 2017 IOP Publishing Ltd.

**Figure 17 nanomaterials-12-02260-f017:**
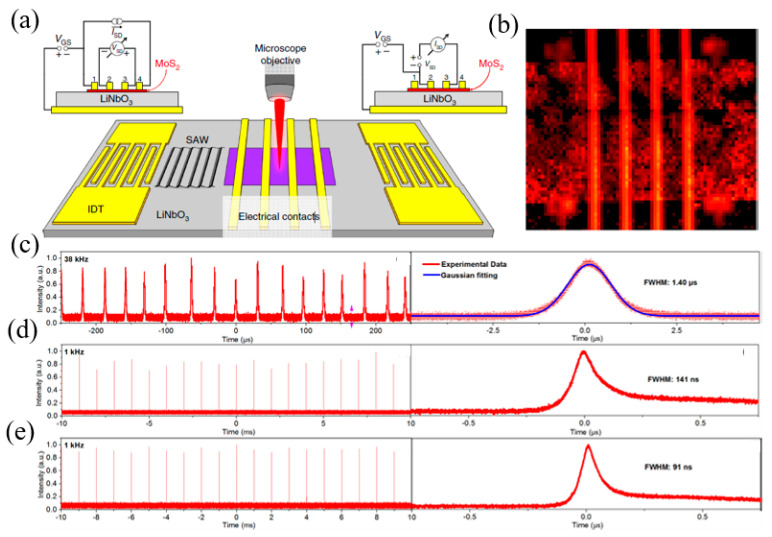
(**a**) Schematic diagram of device based on MoS_2_/LiNbO_3_ combination. (**b**) PL map of the active FET region [292]. Copyright 2015 Nature Publishing Group. (**c**) Pulse trains of lasers and single pulses of WS_2_ Q-switched. (**d**) AOM&WS_2_ Q-switched at 1 kHz. (**e**) AOM&WS_2_ based Q-switching at 1 kHz [293].

**Figure 18 nanomaterials-12-02260-f018:**
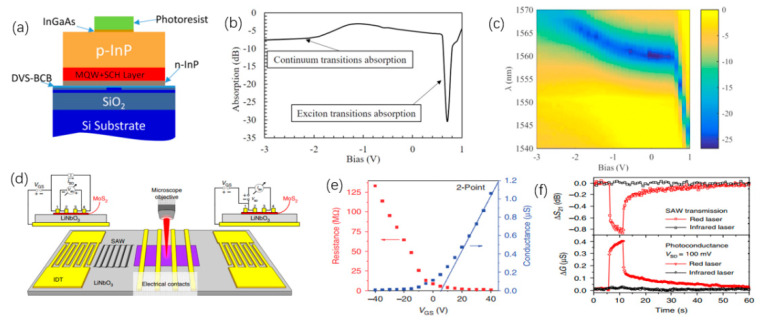
(**a**) Structure diagram of absorption modulator with indium phosphide. (**b**) The absorption curve of the modulator when the bias voltage changes. (**c**) The spectrum of the modulator when the bias voltage changes [295]. Copyright 2016 AIP Publishing LLC. (**d**) Experimental diagram of MoS_2_/LiNbO_3_ acousto-electric device. (**e**) Resistance and conductance change curve with VGS. (**f**) The response time of MoS_2_/LiNbO_3_ under two measurement methods [292]. Copyright 2015 American Publishers Limited.

**Figure 19 nanomaterials-12-02260-f019:**
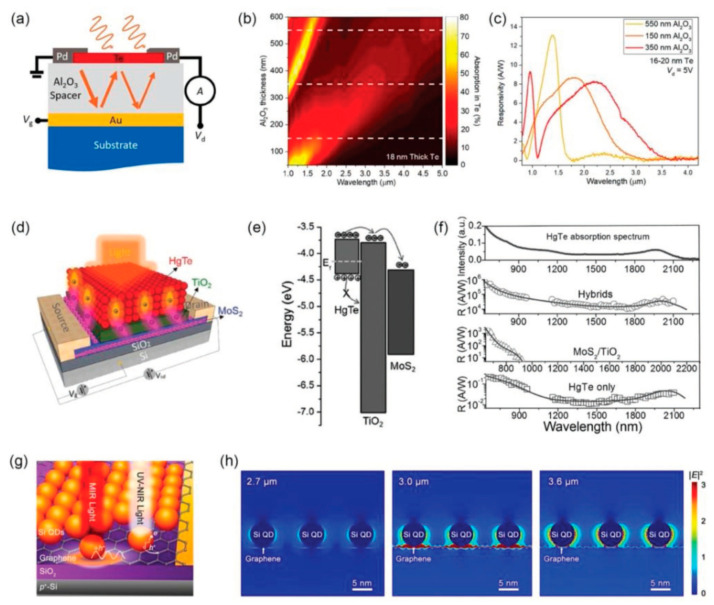
(**a**) Schematic diagram of a 2D Te cavity enhanced photodetector. (**b**) The change of absorption wavelength with the thickness of Al_2_O_3_. (**c**) Spectral response coefficients on Al_2_O_3_ with different thickness [156]. (**d**) Experiment setup of MoS_2_/TiO_2_/HgTe-based hybrid photodetectors. (**e**) Energy band diagrams of MoS_2_–TiO_2_–HgTe hybrid structure. (**f**) Spectral responsivities for HgTe, TiO_2_/MoS_2_, and hybrid devices [297]. Copyright 2017 Wiley-VCH. (**g**) Schematic diagram of the B-doped Si QDs/graphene-based hybrid phototransistor. (**h**) The simulated distribution of the squared electric field |E|^2^ at quantum dots/graphene [301]. Copyright 2017 American Chemical Society.

**Figure 20 nanomaterials-12-02260-f020:**
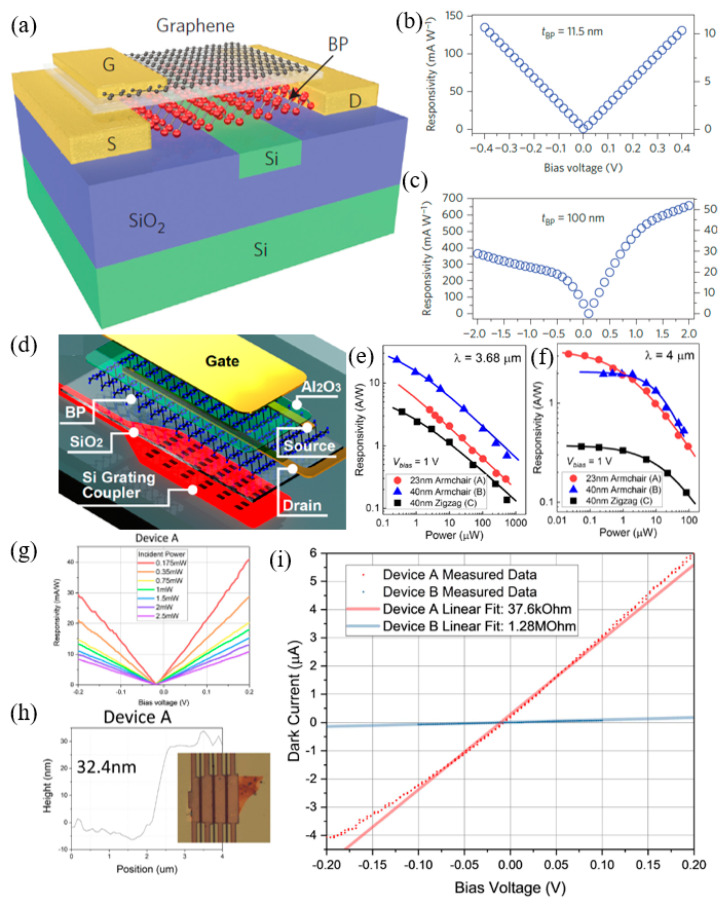
(**a**) 3D demonstration of a BP electric detector with a graphene tip grid. Comparison of the inherent responsivity and internal quantum efficiency (QE) of (**b**) 11.5 nm and (**c**) 100 nm thick BP with application bias. tBP is the thickness of BP [302]. (**d**) An enlarged view of the device output of the system on the waveguide integrated chip with a BP PD. (**e**) The power of both devices depends on the response rate at (**e**) 3.68 μm and (**f**) 4 μm [303]. Copyright 2018 American Chemical Society. (**g**) AFM and microscope images from Device A. (**h**) Measured (dots) and linear fit (lines) dark current and resistance of Devices A. (**i**) The relation between the response rate of Device B and the incident laser power [304]. Copyright 2018 IOP Publishing Ltd.

**Figure 21 nanomaterials-12-02260-f021:**
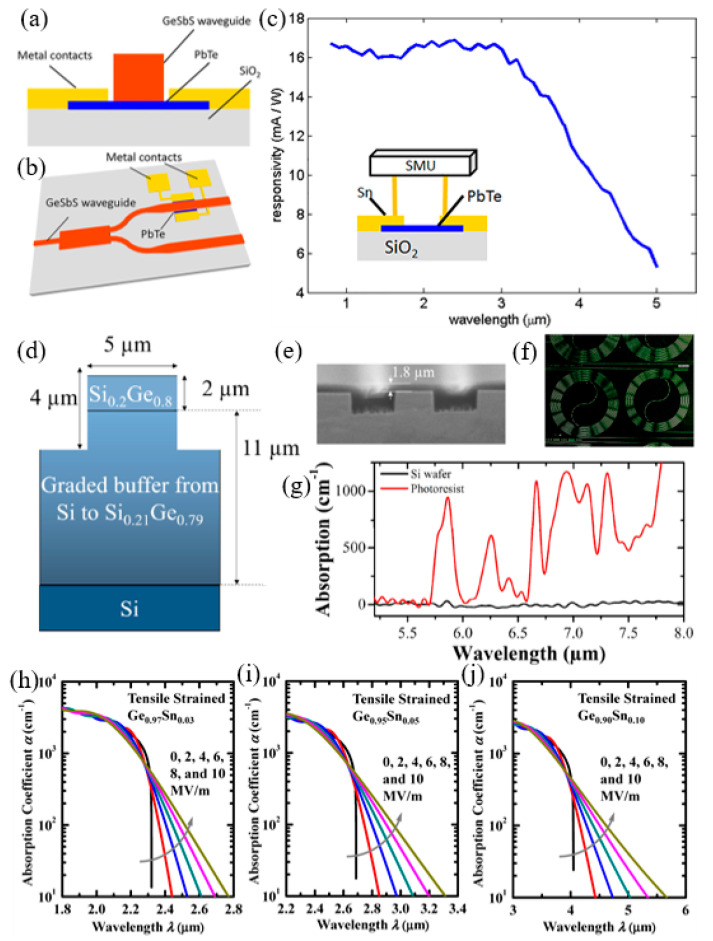
(**a**) Diagram of the cross-section of the integrated MIR detector. (**b**) Schematic diagram of the chip in the device. (**c**) The response of sample 1 at different wavelengths at 213 K with bias voltage of 10 V [307]. Copyright 2016 American Institute of Physics. (**d**) Schematic cross-section of the waveguide considered. (**e**) SEM image of the waveguides covered by S1818 photoresist. (**f**) Top view of the resulting sample. (**g**) Absorption coefficient of the photoresist [308]. Copyright 2018 The Optical Society. (**h**–**j**) Amplitude and wavelength diagrams of Ge_0.97_Sn_0.03_, Ge_0.95_Sn_0.05_ and Ge_0.90_Sn_0.10_ waveguide modulators under different applied electric field conditions [309]. Copyright 2015 The Optical Society.

**Figure 22 nanomaterials-12-02260-f022:**
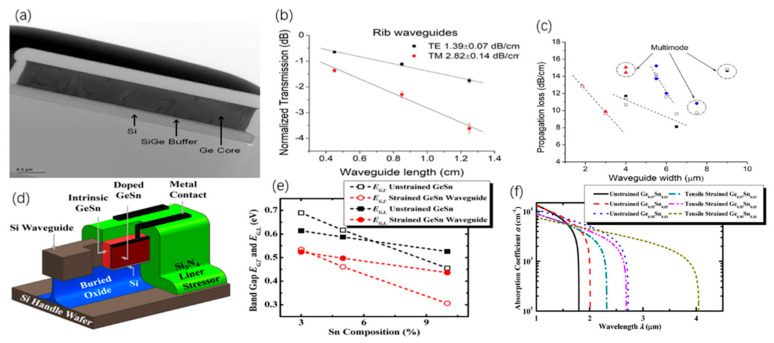
(**a**) Cross-sectional view of Ge-on-SOI microscope. (**b**) The normalized transmission of rib waveguide varies with waveguide length. (**c**) The relationship between the transmission loss and length of the waveguide [309]. Copyright 2018 IOP Publishing Ltd. (**d**) Structure diagram of stretchable GeSn waveguide. (**e**) Band gap change of GeSn in relaxed and stretched states. (**f**) Absorption spectrum when the content of Sn in GeSn changes [308]. Copyright 2015 The Optical Society.

**Figure 23 nanomaterials-12-02260-f023:**
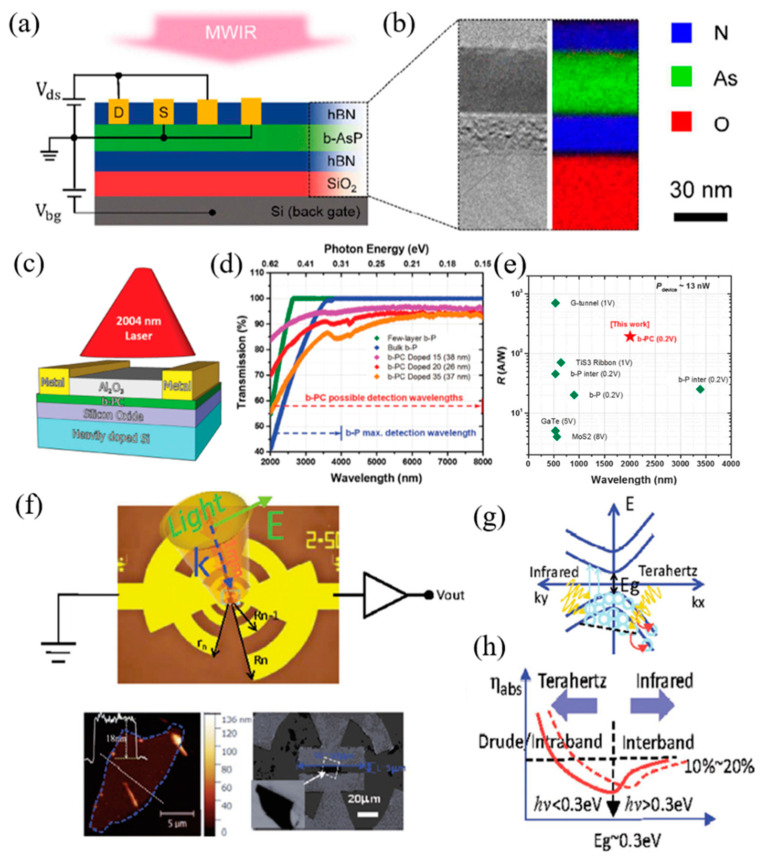
(**a**) Section diagram of the as-fabricated hBN/b-As_0.83_P_0.17_/hBN heterostructure PD. (**b**) Cross section TEM image (Left) and element analysis diagram (right) of the device [100]. (**c**) Schematic diagram of the b-PC phototransistor. (**d**) The AFM of the b-PC phototransistor. (**e**) The response capacity of b-PC is compared to photodetectors reported in recent years measured at the same incident power on the active region [311]. Copyright 2017 Wiley-Blackwell. (**f**) Antenna integrated BP photoconductor. (**g**) The interaction of incident infrared and terahertz photons results in the electron–hole transition of BP slices. (**h**) The boundary between free hole absorption and the generation of inter-band electron-hole pairs is marked by an absorption spectral profile [312]. Copyright 2017 Wiley-VCH Verlag.

**Figure 24 nanomaterials-12-02260-f024:**
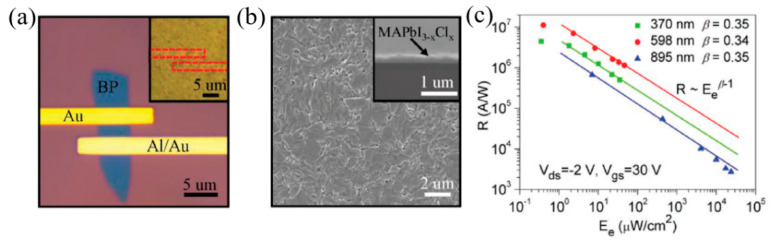
(**a**) Image of a BP/MAPbI_3−X_Cl_X_ Schottky FET. The inset figure exhibits decorated with MAPbI_3−X_Cl_X_ perovskite. (**b**) A typical plane view SEM image of the MAPbI_3−X_Cl_X_ perovskite on silicon substrate. The inset figure shows the corresponding cross-sectional SEM image. (**c**) The response rate increases with the decrease in light intensity [313]. Copyright 2019 Wiley-VCH Verlag.

**Figure 25 nanomaterials-12-02260-f025:**
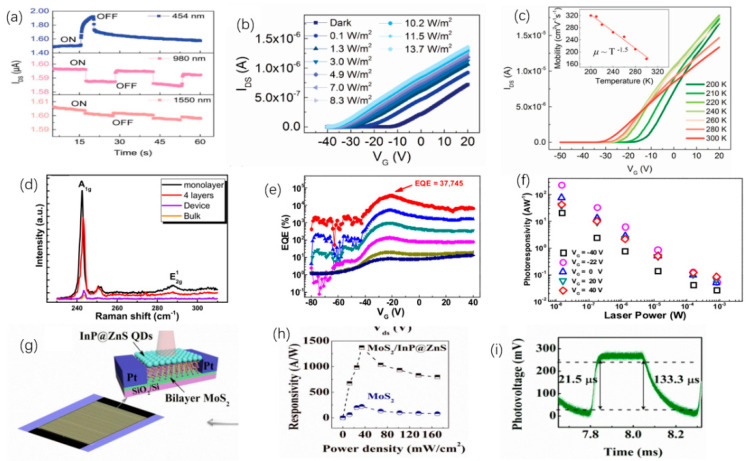
(**a**) Transistor light response excited by light sources in different wavelength bands. (**b**) When the light wavelength is constant, the influence of the ambient light intensity changes to the transistor. (**c**) The influence of temperature changes to transistor in dark environments [314]. Copyright 2018 Wiley-Blackwell. (**d**) Raman spectrum when the number of layers and shape of MoSe_2_ change. (**e**) Graph of the influence of laser power and gate voltage on external electron efficiency. (**f**) The influence of laser power on light response rate [315]. Copyright 2017 IOP Publishing Ltd. (**g**) Structure diagram of MoS_2_ served as a transistor. (**h**) The relationship curve of light response with power density. (**i**) A light response period curve of a transistor [316]. Copyright 2020 American Chemical Society.

**Figure 26 nanomaterials-12-02260-f026:**
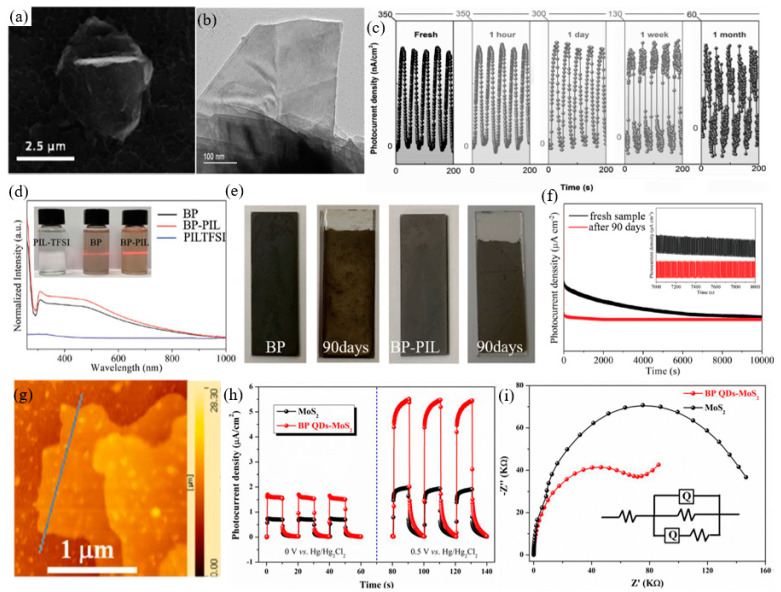
(**a**) Scanning electron microscope photograph of exfoliated BP. (**b**) Transmission electron microscope photograph of exfoliated BP. (**c**) Current density curve with time [317]. Copyright 2017 Wiley-VCH Verlag [318]. Copyright 2019 Wiley-VCH Verlag. (**d**) The optical and composition characteristic analysis. (**e**) The imaging of BP QDs and BP-PIL, which undergo three months testing. (**f**) The photocurrent intensity curves of the newly prepared material and the stability tested material, and the illustration shows the received signal during 7000–8000 s [318]. Copyright 2019 Wiley-VCH Verlag. (**g**) The atomic force microscope image of BP QDs-MoS_2_ compound materials. (**h**) Photocurrent intensity curves of MoS_2_ nanosheets and BP QDS-MOS_2_ compound materials at applied voltages of 0 and 0.5 V in alkaline solution. (**i**) Electrochemical impedance diagrams of MoS_2_ nanosheets and BP QDS-MOS_2_ compound materials at applied voltages of 0 and 0.5 V in alkaline solution [319]. Copyright 2020 Elsevier BV.

**Figure 27 nanomaterials-12-02260-f027:**
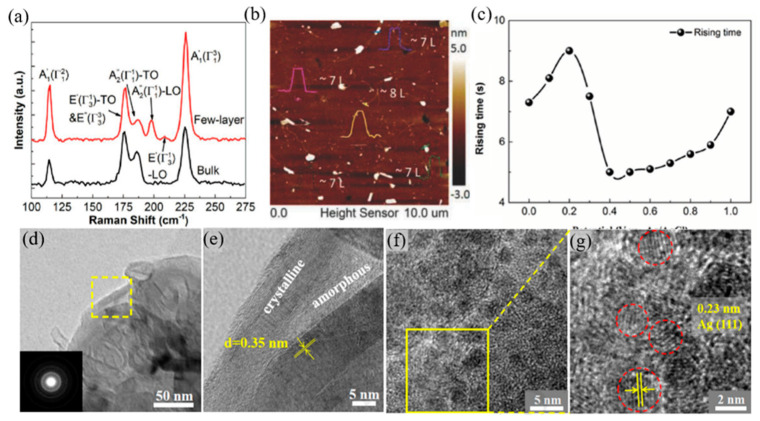
(**a**) Raman atlas of InSe nanosheets after InSe blocks and LPE. (**b**) AFM diagram of InSe nanosheet. (**c**) Response time of PEC type InSe PD under 0–1 V bias in 2 M KOH electrolyte [320]. Copyright 2017 Wiley-VCH Verlag. (**d**) Transmission electron microscope (TEM) photograph of WO_3−x_ materials. (**e**) High resolution TEM photograph of the region in (**d**). (**f**) High resolution TEM photograph of Ag/WO_3−x_ heterostructure. (**g**) High resolution TEM photograph of the region in (**f**) [321]. Copyright 2018 Elsevier.

**Figure 28 nanomaterials-12-02260-f028:**
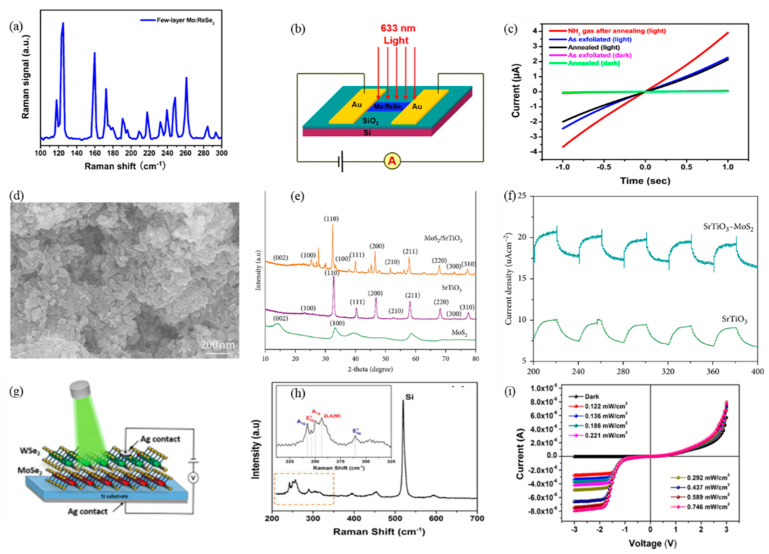
(**a**) Raman spectrum of Mo: ReSe_2_ material. (**b**) Experimental schematic diagram of using Mo: ReSe_2_ as photodetector. (**c**) Voltammetric characteristic diagram of Mo: ReSe_2_ material with different conditions [327]. Copyright 2014 Nature Publishing Group. (**d**) Scanning electron microscope image of SrTiO_3_@ MoS_2_. (**e**) X-ray diffraction images of several materials. (**f**) Comparison chart of current intensity between SrTiO_3_ and SrTiO_3_@ MoS_2_ [123]. Copyright 2020 Hindawi Publishing Corporation. (**g**) Schematic diagram of the MoSe_2_-WSe_2_ structure for light response measurement. (**h**) Raman spectrum of MoSe_2_-WSe_2_. (**i**) Volt-ampere characteristic curve of MoSe_2_-WSe_2_ under different light intensity [328]. Copyright 2020 American Chemical Society.

**Figure 29 nanomaterials-12-02260-f029:**
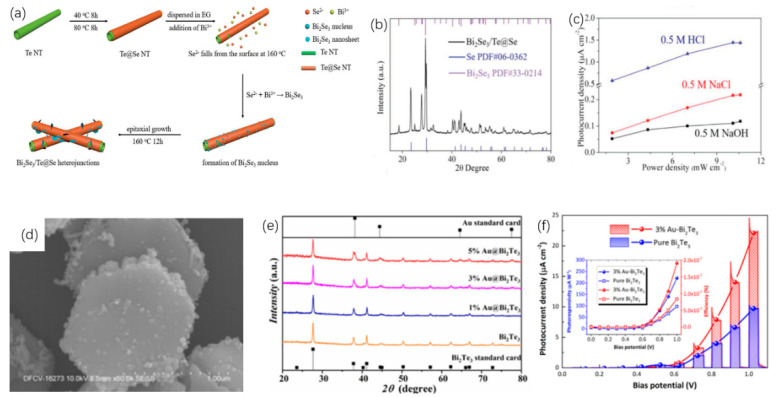
(**a**) Flow chart of Bi_2_Se_3_/Te@Se preparation. (**b**) X-ray diffraction spectra of several materials. (**c**) The light response intensity curve of Bi_2_Se_3_/Te@Se in different alkaline solutions [329]. Copyright2019 Wiley-Blackwell. (**d**) SEM image of 3% Au-Bi_2_Te_3_. (**e**) X-ray diffraction spectra of pure Bi_2_Te_3_ and Au-Bi_2_Te_3_ of different concentrations. (**f**) Curve of photocurrent with the change of bias voltage [331]. Copyright 2020 Elsevier Ltd.

**Figure 30 nanomaterials-12-02260-f030:**
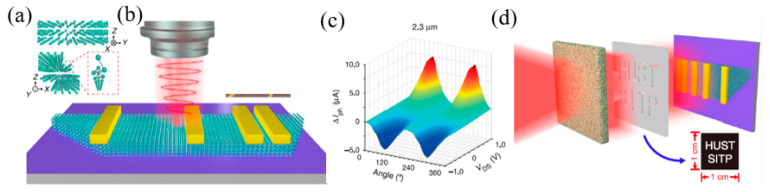
(**a**) The crystal structure imaging of Te, and wherein three atoms form a chain. (**b**) Schematic diagram of Te PD. (**c**) The net polarized photo-generated current ΔIp when the band of incident light is 2.3 μm in ambient temperature, and the incident power is 6.0 mW. (**d**) Infrared imaging method structure diagram [160]. Copyright 2020 Nature Publishing Group.

**Table 2 nanomaterials-12-02260-t002:** The 2D material candidates for LEDs operation.

2D Materials	Bandgap (eV)	Wavelength (μm)	Luminous Mode	Ref.
BP	0.3~1.5	0.83~4.13	photoluminescence	[185]
2D Ruddlesden–Popper perovskites	1.53	0.81	photoluminescence	[198]
GDY	0.46~1.10	1.12~2.69	photoluminescence	[141]
Er-doped MoS_2_	0.8	1.55	electroluminescence	[190]
WSe_2_	1.0~2.4	0.51~1.24	electroluminescence	[201]

**Table 3 nanomaterials-12-02260-t003:** The laser operation based on 2D materials.

2D Materials	Wavelength (μm)	Pulse Duration	Threshold	Frequency	Ref.
BP	2.97~3	CW	302.6 mW	12.43 KHz	[205]
BP	3~5	42 ps	613 mW	24 MHz	[212]
Bi_2_Te_3_ deposited on CaF_2_	2.979	1.37 μs	3.39 μJ	81.96 KHz	[210]
MoS_2_	2.8	ND	430 mW	62.42 KHz	[213]
Bi_2_Te_3_	2.8~3.0	ND	3.0~3.5 W	ND	[210]
Bi_2_Se_3_	3.0	1.5μs	48 mW	55.1 KHz	[216]

**Table 4 nanomaterials-12-02260-t004:** The 2D material candidates for LEDs operation.

2D Materials	Spectral Range (μm)	Room Temperature Responsivity	On/Off Ratio	Specific Detectivity (Jones)	Ref.
Few-layer Te	1.4~3.5	13 and 8 A W^−1^ at 1400 and 2400 nm	10^5^	2 × 10^9^ at 1700 nm	[156]
b-As_0.83_P_0.17_	2.4~8.05	15~30 A W^−1^	ND	10^8^	[331]
BN/Multilayer b-As_0.83_P_0.17_/BN	3.4 ~7.7	1.2 mA W^−1^ at 7700 nm	110	ND	[100]
PdSe_2_	0.45~10.6	42.1 A W^−1^ at 10600 nm	ND	0.28	[119,126]

## Data Availability

Not applicable.

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
