# Peer review of "Mid-Infrared Optoelectronic Devices Based on Two-Dimensional Materials beyond Graphene: Status and Trends"

_nanomaterials, 2022, doi:10.3390/nano12132260_

Round 1
Reviewer 1 Report
This is an interesting and detailed review on the application of 2D materials on Mid-infrared optoelectronics
I suggest publishing the paper after some revisions
· Can the author comment about the Environmental stability of Black Phosphorous and of Transition metal dichalcogenides with Selenide and Telluride content?
· TMD have gap ranging from Visible to NIR (1-2 eV). Can the author better specify the spectral range of operation of such materials?
· A summary scheme of 2D materials with their bandgap and spectral region of operation would help the reader the better classify the different materials.
· Regarding the emission properties 2D materials (3.1 Spontaneous emission) I would suggest a clearer distinction between the emission from the materials and the emission from defect in the material such as single photon sources. For example, h-BN does not show emission (in the VIS) but can support local emitting defect.
· In the paragraph on laser operation, I would in detail comment about (with a table) the pulse duration of the several lasers and their threshold
· As far as the LED operation the author must add more details about charge injection scheme. Moreover, what 2D materials are well suited for LED operation?
· What is the Photodetection spectral range of the several type of 2D materials?
Author Response
Revisions and list of changes (nanomaterials-1738091).
Replies to the 1st reviewer's comments (nanomaterials-1738091).
Comment 1. The authors review the recent progress of semiconducting 2D materials in MIR optoelectronic devices that present the suitable category 2D materials for light emission devices, modulators, and photodetectors in the MIR band. However, there are some problems left; thereby, I suggest this manuscript can be accepted for publication after minor revision. The following lists are some specific comments about the misleading points in this manuscript that might need further clarification:
With ultrahigh mobility and stability, graphene possesses stronger optical absorption than some 2D narrow-bandgap semiconductors such as BP at IR wavelengths (Nature Photonics, 2015, 9, 247-252 Appl. Phys. Lett., 2012, 101, 111110). Moreover, optoelectronic devices based on graphene have shown obvious advantages (ACS Nano, 2021, 15, 10084-10094, Light Science Applications, 2020, 9, 29). Therefore, the authors need to illustrate the reasons for MIR optoelectronic devices based on 2D semiconductors in the abstract and introduction.
Reply 1: Thank you very much for your good suggestion. According to your suggestions, we have added some content to illustrate the reasons for MIR optoelectronic devices based on 2D semiconductors in the both abstract and introduction sections and marked as blue color text in the revised manuscript. Additionally, all these mentioned references are cited in this revised manuscript.
Comment 2. In the “Light emission devices” section, Figure 2c, Figure 3 and Figure 4 are not related to MIR, and 2D-Ruddlesden-Popper Perovskites, GDY and TMDCs are difficult for light sources at MIR.
Reply 2: Thank you very much for your good suggestion. According to your suggestion, firstly, in the “Light emission devices” section, Figure 2c, Figure 3 and Figure 4 have been replaced with some content relate MIR emission applications. Secondly, 2D-Ruddlesden-Popper Perovskites, GDY and TMDCs are not suitable for MIR emission applications, and the corresponding content has been deleted.
Comment 3. In the “Modulator” section, the authors had better delete some Figures of modulators at visible wavelengths and NIR and introduce the detailed development of modulators at MIR. Moreover, the thermo-optic effect is important for practical applications, which should also be reviewed in this manuscript.
Reply 3: Thank you very much for your good suggestion. According to your suggestion, in the “Modulator” section, the figures relate to visible and NIR band have been deleted. The detailed development of modulators at MIR is added in the beginning of section 3, and marked as blue color text in the revised manuscript. Moreover, we have added some content about thermo-optic approach at the end of section “4.3 Other approaches” to introduce this important technique in the revised manuscript. Moreover, most reported investigations of thermo-optic approach are concentrated on visible to NIR band, thus, we only introduce the underlying operation mechanism and potential 2D material candidates of MIR thermo-optic modulators, and all these added content is marked as blue color text in the revised manuscript.
“Thermo-optic modulation is another commonly used effective approach to realize high performance modulator devices, which is rely on temperature dependent refractive index change of a certain material. Generally, the refractive index of materials is varied through heating process. Then, the incident light pass through the employed optical waveguide filled with 2D materials, and the phase modulation via a resonant structure or interferometry can be achieved. As aforementioned, 2D materials possess superior properties, such as high electrical, thermal conductivity, wide optical absorption region, which enable 2D materials hold great potential for thermo-optic modulation applications. However, most reported investigations of thermo-optic approach are concentrated on visible to NIR band. To meet the requirements of different applications, it is of great significance to development MIR thermo-optic modulators.”
Comment 4. In the “Results Conclusion and outlook” section, the authors are expected to conclude the advantages of 2D materials for MIR devices, discuss the remaining problems of MIR devices based on 2D materials, and illustrate the possibility of 2D materials for practical applications at MIR.
Reply 4: Thank you very much for your good suggestion. According to your suggestion, the “Results Conclusion and outlook” section has been re-written, which focus on the advantages of 2D materials for MIR devices, discuss the remaining problems of MIR devices based on 2D materials, and illustrate the possibility of 2D materials for practical applications at MIR and marked as blue color text in the revised manuscript.
- Results Conclusion and outlook
Since the extraordinary optoelectronic properties, 2D materials beyond graphene are intensively employed to fabricate MIR optoelectronic devices. In this review, we investi-gated the recent progress of 2D materials-based MIR optoelectronic devices, including 2D material candidates are suitable for MIR applications, such as BP, TMDCs, GDY, 2D Te nanoflakes, perovskites, and Topological insulators. As well as their device application in MIR band, in particular, light emission devices, modulators and photodetectors.
- In terms of LEDs, compared to commercial available LEDs, the external quantum efficiency, operation stability of 2D materials based LEDs are still too low. To satisfy the demand of practical applications, the external quantum efficiency and operation stability need to be significantly improved. Furthermore, the emitting wavelength needs to be fur-ther extended, which are far from practical applications. For driven mode, more attention should be concentrated on electrically driven mode. Thus, it is of great significance to de-velop novel 2D materials based LEDs with various configurations in the future.
- Regarding single-photon emitters based on 2D materials, which are thought to be originated from the defects. However, the underlying physical mechanism, excitation processes and atomic structure are still under debate. Meanwhile, the emitting wavelength needs to be extended into deep MIR region.
- The lasing threshold is relatively low at a lower temperature, however, for room temperature lasing, the threshold needs to be significantly suppressed.
- For modulator applications, the irradiation damage threshold of 2D materials need to be improved. In addition, the long term operation stability needs to be enhanced as well, which plays a determine role for practical applications. Moreover, only few kinds of 2D material can be utilized for electro-optic modulators. Further improving the architecture of optical modulators and exploring novel 2D materials may provide an alternative means to overcome these challenges.
- In terms of MIR photodetectors, since the weaker optical absorption of MIR light, the photo responsivity, carrier mobility and response speed are much lower and slower than that of visible and NIR photodetectors. Combined with other materials with higher MIR light absorption coefficient to establish heterojunction may provide an effective way to solve these problems.
- For MIR imaging applications, the recent devices demonstrate mainly to adopt a point photodetector, which is far from practical applications. MIR imaging devices based on large scale 2D material arrays should be developed.
In conclusion, 2D materials beyond graphene-based MIR optoelectronic devices have already achieved some milestone achievements. However, these devices also face some severe challenges as aforementioned. With continuous investigation, we believe that a more comprehensive understanding of MIR optoelectronic devices based on 2D materials beyond graphene will emerge in the near future as a result of these ongoing concerted re-search efforts.
Comment 5. From my viewpoint, the authors should focus on the devices at MIR and delete the devices at other wavelengths.
Reply 5: Thank you very much for your good suggestion. According to your suggestion, this manuscript has been corrected mainly focus on the MIR band, and devices at other wavelengths has been deleted.

Reviewer 2 Report
This is a clear, up-to-date and well-written review manuscript dealing with the issue of “Mid-Infrared Optoelectronic Devices Based on Two-dimensional Materials Beyond Graphene”.
Overall, this is a very useful reference review for anyone who has to do with the research topic on MIR optoelectronic devices but also for those who are preparing to tackle this research area.
Authors should be congratulated for conducting this work useful for scientist as well as for reference purposes.
Therefore, the content of the manuscript have value for publication.
Specific comments:
p. 40, sect 6 Result and Discussion: The first three lines should be deleted.
Some figures and tables need to be improved for readability….however this is a publisher task
Author Response
Replies to the 2nd reviewer's comments (nanomaterials-1738091).
Comment 1. This is a clear, up-to-date and well-written review manuscript dealing with the issue of “Mid-Infrared Optoelectronic Devices Based on Two-dimensional Materials Beyond Graphene”. Overall, this is a very useful reference review for anyone who has to do with the research topic on MIR optoelectronic devices but also for those who are preparing to tackle this research area. Authors should be congratulated for conducting this work useful for scientist as well as for reference purposes. Therefore, the content of the manuscript have value for publication.
Specific comments:
- 40, sect 6 Result and Discussion: The first three lines should be deleted.
Reply 1: Thank you very much for your good suggestion. According to your suggestion, the first three lines of section 6 have been deleted.
Comment 2. Some figures and tables need to be improved for readability….however this is a publisher task.
Reply 2: Thank you very much for your good suggestion. All figures used in this manuscript have been re-edited and the resolution has been improved.

Reviewer 3 Report
Comments to the Author
The authors review the recent progress of semiconducting 2D materials in MIR optoelectronic devices that present the suitable category 2D materials for light emission devices, modulators, and photodetectors in the MIR band. However, there are some problems left; thereby, I suggest this manuscript can be accepted for publication after minor revision. The following lists are some specific comments about the misleading points in this manuscript that might need further clarification:
1. With ultrahigh mobility and stability, graphene possesses stronger optical absorption than some 2D narrow-bandgap semiconductors such as BP at IR wavelengths (Nature Photonics, 2015, 9, 247-252 Appl. Phys. Lett., 2012, 101, 111110). Moreover, optoelectronic devices based on graphene have shown obvious advantages (ACS Nano, 2021, 15, 10084-10094, Light Science Applications, 2020, 9, 29). Therefore, the authors need to illustrate the reasons for MIR optoelectronic devices based on 2D semiconductors in the abstract and introduction.
2. In the “Light emission devices” section, Figure 2c, Figure 3 and Figure 4 are not related to MIR, and 2D-Ruddlesden-Popper Perovskites, GDY and TMDCs are difficult for light sources at MIR.
3. In the “Modulator” section, the authors had better delete some Figures of modulators at visible wavelengths and NIR and introduce the detailed development of modulators at MIR. Moreover, the thermo-optic effect is important for practical applications, which should also be reviewed in this manuscript.
4. In the “Results Conclusion and outlook” section, the authors are expected to conclude the advantages of 2D materials for MIR devices, discuss the remaining problems of MIR devices based on 2D materials, and illustrate the possibility of 2D materials for practical applications at MIR.
5. From my viewpoint, the authors should focus on the devices at MIR and delete the devices at other wavelengths.
Author Response
Replies to the 3rd reviewer's comments (nanomaterials-1738091).
Comment 1. This is an interesting and detailed review on the application of 2D materials on Mid-infrared optoelectronics. I suggest publishing the paper after some revisions.
Can the author comment about the Environmental stability of Black Phosphorous and of Transition metal dichalcogenides with Selenide and Telluride content?
Reply 1: Thank you very much for your good suggestion. According to your suggestion, the environmental stability of Black Phosphorous and of Transition metal dichalcogenides with Selenide and Telluride content has been added in the section 2.1 and 2.2, respectively. All the revised content is marked as blue color text in the revised manuscript.
For BP:
“However, the instability issues of BP flakes, caused by the oxygen, water and light induced oxidation, are more and more serious, which severely hinder its further development in both academic and practical applications. To solve this issue, tremendous efforts have been employed to enhance the environmental stability of BP flakes, such as encapsulation, covalent/noncovalent functionalization and metal/non-metal modification.”
For TMDCs:
“Additionally, most TMDCs possess stable 1 T phase, which enable TMDCs possess excel-lent environment stability, which is much better than that of BP flakes and comparable to that of 2D tellurium (Te).”
Comment 2: TMDCs have gap ranging from Visible to NIR (1-2 eV). Can the author better specify the spectral range of operation of such materials?
Reply 2: Thank you very much for your good suggestion. According to your suggestion, the spectral range of operation of some representative materials in this manuscript has been summarized in table 1, which is located at the end of section 2 and marked as blue color text in the revised manuscript.
Table 1 the spectral range of operation of some representative materials
|
2D materials |
Bandgap (eV) |
the spectral range of operation (μm) |
Ref. |
|
BP |
0.3 ~ 1.5 |
0.83 ~ 4.13 |
185 |
|
b-AsP |
0.15 |
8.27 |
94 |
|
b-PC |
0.59 |
2.10 |
95 |
|
ReSe2 |
1.1 ~ 1.58 |
0.78 ~ 1.12 |
110 |
|
PdSe2 |
1.3 |
0.95 |
121 |
|
PtSe2 |
0.3 |
4.13 |
116 |
|
GDY |
0.46 ~ 1.10 |
1.12 ~ 2.69 |
141 |
|
2D Te |
0.35 ~ 1.265 |
0.98 ~ 3.54 |
154 |
|
Bi2Te3 |
0.21 |
5.90 |
161 |
|
Sb2Te3 |
0.45 |
2.75 |
164 |
Comment 3: A summary scheme of 2D materials with their bandgap and spectral region of operation would help the reader the better classify the different materials.
Reply 3: Thank you very much for your good suggestion. According to your suggestion, the spectral range of operation of some representative materials in this manuscript has been summarized in table 1, which is located at the end of section 2 and marked as blue color text in the revised manuscript.
Comment 4:·Regarding the emission properties 2D materials (3.1 Spontaneous emission) I would suggest a clearer distinction between the emission from the materials and the emission from defect in the material such as single photon sources. For example, h-BN does not show emission (in the VIS) but can support local emitting defect.
Reply 4: Thank you very much for your good suggestion. According to your suggestions, section 3.1 has been re-organized with spontaneous emission from the materials and the emission from defect in the material, and marked as blue color text in the revised manuscript.
“Recently, 2D TMDCs, including MoSe2, MoS2 and WSe2, due to their sharp linewidth emissions on the lower energy side of the delocalized exciton emission, neutral excitons trapped at anisotropic confining potentials from defects, are suitable for single photon emitter applications. Meanwhile, combined with the plasmonic effect and the strain engi-neering, the single photon emitter performance can be further enhanced. The single pho-ton emission was verified to originate from spatially localized regions of the TMDCs sam-ples. These results suggest that single photon infrared emitters can be realized in 2D ma-terials and have promising applications in quantum devices. Motived by these, …”
Comment 5: In the paragraph on laser operation, I would in detail comment about (with a table) the pulse duration of the several lasers and their threshold.
Reply 5: Thank you very much for your good suggestion. According to your suggestion, the laser operation based on 2D materials has been summarized in table 3, which is located at the end of section 3 and marked as blue color text in the revised manuscript.
Table 3 the laser operation based on 2D materials
|
2D materials |
Wavelength (μm) |
Pulse duration |
Threshold |
Frequency |
Ref. |
|
BP |
2.97 ~ 3 |
CW |
302.6 mW |
12.43 KHz |
205 |
|
BP |
3 ~ 5 |
42 ps |
613 mW |
24 MHz |
212 |
|
Bi2Te3 deposited on CaF2 |
2.979 |
1.37 μs |
3.39 μJ |
81.96 KHz |
210 |
|
MoS2 |
2.8 |
ND |
430 mW |
62.42 KHz |
213 |
|
Bi2Te3 |
2.8 ~ 3.0 |
ND |
3.0 ~ 3.5 W |
ND |
216 |
|
Bi2Se3 |
3.0 |
1.5μs |
48 mW |
55.1 KHz |
217 |
Comment 6: As far as the LED operation the author must add more details about charge injection scheme. Moreover, what 2D materials are well suited for LED operation?
Reply 6: Thank you very much for your good suggestion. According to your suggestion, firstly, we have added a paragraph to describe about the charge injection scheme as “Regarding LEDs applications, 2D materials can be introduced as interlayers in LEDs, in-cluding anode, HTL (hole transport layer), HIL (hole injection layer) and EIL (electron transport layer), and the performance of device can be significantly enhanced due to the improved work function, effective electron blocking, increased the hole injection from an-ode into the organic layer, which enable more holes and electrons recombining in emis-sion layers. Moreover, the electron injection can be enhanced as well when 2D materials are employed as dopant in EIL.” Secondly, 2D materials candidates suitable for LED operation are summarized in table 2, which locate at Page. 9.
Table 2 2D material candidates for LEDs operation
|
2D materials |
Bandgap (eV) |
Wavelength (μm) |
luminous mode |
Ref. |
|
BP |
0.3 ~ 1.5 |
0.83 ~ 4.13 |
photoluminescence |
185 |
|
2D-Ruddlesden-Popper perovskites |
1.53 |
0.81 |
photoluminescence |
198 |
|
GDY |
0.46 ~ 1.10 |
1.12 ~ 2.69 |
photoluminescence |
141 |
|
Er-doped MoS2 |
0.8 |
1.55 |
electroluminescence |
190 |
|
WSe2 |
1.0 ~ 2.4 |
0.51 ~ 1.24 |
electroluminescence |
200 |
|
MoTe2 |
1.1 |
1.127 |
electroluminescence |
262 |
Comment 7: What is the Photodetection spectral range of the several type of 2D materials?
Reply 7: Thank you very much for your good suggestion. According to your suggestion, the photodetection spectral range of the several type of 2D materials has been summarized in table 4, which is located at the end of section 3 and marked as blue color text in the revised manuscript.
Table 4. 2D material candidates for LEDs operation
|
2D materials |
Spectral range (μm) |
Room‑temperature responsivity |
On/off ratio |
Specific detectivity (Jones) |
Ref. |
|
Few-layer Te |
1.4 ~ 3.5 |
13 and 8 A W-1 at 1400 and 2400 nm |
105 |
2 × 109 at 1700 nm |
156 |
|
b-As0.83P0.17 |
2.4 ~ 8.05 |
15 ~ 30 A W-1 |
ND |
108 |
334 |
|
BN/Multilayer b-As0.83P0.17/BN |
3.4 ~7.7 |
1.2 mA W-1 at 7700 nm |
110 |
ND |
100 |
|
PdSe2 |
0.45 ~ 10.6 |
42.1 A W-1 at 10600 nm |
ND |
0.28 |
119, 126 |
